# Stochastic optimal open-loop control as a theory of force and impedance planning via muscle co-contraction

**Bastien Berret**[1,2,3]\*, **Frédéric Jean**[4]

**1** Université Paris-Saclay CIAMS, Orsay, France, **2** CIAMS, Université d'Orléans, Orléans, France, **3** Institut Universitaire de France, Paris, France, **4** Unité de Mathématiques Appliquées, ENSTA Paris, Institut Polytechnique de Paris, Palaiseau, France

\* bastien.berret@universite-paris-saclay.fr

**Data Availability Statement:** There is no new experimental data in this study. However, the Matlab codes used to produce the simulations are available from http://hebergement.universite-paris-saclay.fr/berret/software/SOOC.zip. Note that

## Abstract

Understanding the underpinnings of biological motor control is an important issue in movement neuroscience. Optimal control theory is a leading framework to rationalize this problem in computational terms. Previously, optimal control models have been devised either in deterministic or in stochastic settings to account for different aspects of motor control (e.g. average behavior versus trial-to-trial variability). While these approaches have yielded valuable insights about motor control, they typically fail in explaining muscle co-contraction. Co-contraction of a group of muscles associated to a motor function (e.g. agonist and antagonist muscles spanning a joint) contributes to modulate the mechanical impedance of the neuromusculoskeletal system (e.g. joint viscoelasticity) and is thought to be mainly under the influence of descending signals from the brain. Here we present a theory suggesting that one primary goal of motor planning may be to issue feedforward (open-loop) motor commands that optimally specify both force and impedance, according to noisy neuromusculoskeletal dynamics and to optimality criteria based on effort and variance. We show that the proposed framework naturally accounts for several previous experimental findings regarding the regulation of force and impedance via muscle co-contraction in the upper-limb. Stochastic optimal (closed-loop) control, preprogramming feedback gains but requiring on-line state estimation processes through long-latency sensory feedback loops, may then complement this nominal feedforward motor command to fully determine the limb's mechanical impedance. The proposed stochastic optimal open-loop control theory may provide new insights about the general articulation of feedforward/feedback control mechanisms and justify the occurrence of muscle co-contraction in the neural control of movement.

## Author summary

This study presents a novel computational theory to explain the planning of force and impedance (e.g. viscoelasticity) in the neural control of movement. It assumes that one main goal of motor planning is to elaborate feedforward motor commands that determine both the force and the impedance required for the task at hand. These feedforward motor

the codes depend on a third-party software (GPOPS).

**Funding:** This work was partly supported by a public grant overseen by the French National Research Agency as part of the Investissement d'Avenir program, through the iCODE Institute project funded by the IDEX Paris-Saclay, ANR-11-IDEX-0003-02. The funders had no role in study design, data collection and analysis, decision to publish, or preparation of the manuscript.

**Competing interests:** The authors have declared that no competing interests exist.

commands (i.e. that are defined prior to movement execution) are designed to minimize effort and variance costs considering the uncertainty arising from sensorimotor or environmental noise. A major outcome of this mathematical framework is the explanation of muscle co-contraction (i.e. the concurrent contraction of a group of muscles involved in a motor function). Muscle co-contraction has been shown to occur in many situations but previous modeling works struggled to account for it. Although effortful, co-contraction contributes to increase the robustness of motor behavior (e.g. small variance) upstream of sophisticated optimal closed-loop control processes that require state estimation from delayed sensory feedback to function. This work may have implications regarding our understanding of the neural control of movement in computational terms. It also provides a theoretical ground to explain how to optimally plan force and impedance within a general and versatile framework.

## Introduction

Optimal control theory is a leading framework for understanding biological motor behavior in computational terms [1–4]. Historically, this research has been carried out along two lines. On the one hand, deterministic optimal control (DOC) theory focused on the planning stage and sought to explain average motor behaviors in humans or animals. The minimum jerk and minimum torque change models are well-known representatives of this line of research [5, 6], which provided researchers with simple models accounting for the formation of average trajectories (e.g. bell-shaped velocity profiles in reaching tasks). This laid the foundations for more advanced studies like inverse optimal control ones, where the goal is to recover relevant optimality criteria from (averaged) experimental motion data [7, 8]. On the other hand, stochastic optimal control (SOC) theory was used to account for the variability of biological movement observed across multiple repetitions of the same task [9–11]. The noise that affects the neuro-musculoskeletal system, and the uncertainty it induces about movement performance, are taken into account in this approach [12, 13]. This class of model can also be used to explain motor planning (e.g. via the specification of feedback gains prior to movement onset) but the genuine motor commands are only revealed along the course of the movement, once the current state of the system has been optimally estimated (e.g. hand/joint positions, velocities etc.). The SOC theory led to a number of valuable predictions among which the minimal intervention principle, stating that errors are corrected on-line only when they affect the goal of the task [9].

However, both of these approaches fail at accounting for a fundamental motor control strategy used by the central nervous system (CNS) and often referred to as co-contraction or co-activation of muscles groups (see [14] for a recent review). This frequent phenomenon is known since more than a century and the work of Demenÿ [15], and has been investigated extensively since then. There is now a strong evidence that co-contraction is voluntarily used by the CNS in a number of tasks, especially those requiring high stability, robustness or end-point accuracy [16–18]. Co-contraction indeed contributes to modulate the mechanical impedance of the neuromusculoskeletal system. For instance, co-contraction can drastically increase the apparent joint stiffness by a factor 4 to 7 [19]. This effect results both from the summation of intrinsic stiffness of muscles around a common joint [20, 21] and reflexes [22–24]. The former short-range stiffness implements an instantaneous (feedback-free) mechanism. The latter implements both a short-latency (low-level) feedback mechanism via fast-conducting mono- or oligo-synaptic spinal pathways [response latency at muscle level ~20-40 ms

after a mechanical perturbation] and a long-latency (high-level) feedback mechanism via transcortical pathways [response latency ~50-100 ms]. The two above-mentioned approaches (DOC and SOC) are not able to account for co-contraction in a principled way for fundamentally distinct reasons. First, co-contraction contributes to modulate the effective limb's impedance (e.g. joint viscoelasticity), whose actual effect can only be seen when unexpected perturbations are applied to the limb [19, 25]. As there are no such random perturbations in deterministic models, they will usually not predict co-contraction. Indeed, there is no functional gain at co-contracting opposing muscles in those models. Co-contraction just appears as a waste of energy considering that such models typically aim at minimizing effort or energy-like costs [26, 27]. Therefore, whenever a deterministic model exhibits co-contraction, it is an artifact of muscle modeling (e.g. due to response times of muscle activation dynamics) that does not serve any task-oriented, functional purpose. In SOC models, the presence of sensorimotor and environmental noise is taken into account so that co-contraction could become a relevant strategy regarding disturbance rejection and task achievement. However, SOC controllers typically exhibit reciprocal muscle activation patterns on average because they also minimize (expected) effort costs (e.g. see Fig. 2 in [28] or Fig. 3a in [29]), and correct errors using sensory feedback and reciprocal activations that are less costly than co-contraction. A few studies have nevertheless attempted to predict co-contraction from the SOC framework. These studies had to rely on advanced noise models explicitly reducing signal-dependent variance during co-contraction or on advanced viscoelastic muscle models yielding co-contraction without clear task dependency or functional purpose [29, 30]. More fundamentally, a closed-loop optimal control scheme requires optimal state estimation combining delayed sensory signals with an efferent copy of the motor command –the latter being converted into state variables via forward internal models– [31]. The neural substrate underlying SOC is thought to involve the long-latency transcortical pathway passing through the primary motor cortex [32–34]. This may seem to contrast with the feedforward nature of impedance that has been demonstrated in several studies [16, 18, 35–37]. However, the planning of optimal feedback gains may be viewed as a form of feedforward control of impedance in SOC models [38]. The main difference with co-contraction is that control via feedback gains critically depends on the ability of the CNS to form accurate estimates of the current system state. As this ability may be limited in some cases (e.g. unpredictable interaction with the environment, unstable task or too fast motion), co-contraction may constitute an alternative strategy to regulate mechanical impedance without the involvement of high-level, long-latency feedback mechanisms. In this vein, several studies on deafferented monkeys (without feedback circuitry at all) suggested that an equilibrium point/trajectory resulting from the co-contraction of opposing muscles was preprogrammed by the CNS during point-to-point movements without sight of the arm [39–42]. Similar conclusions were drawn with deafferented patients who were able to perform relatively accurate reaching movements without on-line vision –if allowed to see their arm transiently prior to movement execution– [43]. Furthermore, neurophysiological studies seem to agree that muscle co-contraction has a central origin with little contribution from spinal mechanisms [14, 19, 44]. Noticeably, during co-contraction of antagonistic muscles, disynaptic reciprocal inhibition has been shown to be reduced by central signals [45, 46]. This highlights the singularity of muscle co-contraction in impedance control and departs from the reciprocal activations predicted by standard models based on DOC or SOC theories. For these reasons, co-contraction may be a critical feature of descending motor commands (i.e. *open-loop control* in computational terms) which may serve to generate stable motor behaviors ahead of the optimal closed-loop control machinery.

In this paper, we thus propose a novel stochastic optimal control framework to determine force and impedance –via muscle co-contraction– at the stage of motor planning. Our

approach lies in-between DOC and SOC theories from a mathematical standpoint and we refer to it as *stochastic optimal open-loop control* (SOOC) theory to stress that we consider stochastic dynamical systems controlled by open-loop, deterministic controls [47]. This work is in the vein of seminal motor control studies [5, 48, 49]. We generalize and extend these approaches to the planning of upper-limb movement within a versatile mathematical framework that can handle a variety of motor tasks, types of noise, nonlinear limb dynamics and cost functions. The proposed theory primarily accounts for co-contraction as a means to modulate the apparent mechanical impedance of the musculoskeletal system via feedforward, descending motor commands that do not require any advanced on-line estimation of the system state. Although we use the term *open-loop* –in the sense of control theory– we do not necessarily exclude the role of reflexes that contribute to the spring-like behavior of intact muscles beyond their short-range stiffness. However, we do exclude from this open-loop terminology all the optimal closed-loop control processes integrating sensory data during movement execution through transcortical feedback loops [32]. The critical role of SOC is rather attributed to those long-latency, sophisticated and task-dependent motor responses that are triggered by the CNS to correct large-enough external perturbations [33, 34].

## Materials and methods

Our working hypothesis is that both force and mechanical impedance are planned by the brain via descending motor commands. To illustrate our purpose, we will focus on the control of arm posture and movement, and compare the predictions made by our framework to existing experimental data. In this work, the major premise is to assume open-loop control (which makes sense at the stage of the motor planning process) while acknowledging the stochastic nature of the neuromusculoskeletal system. We shall illustrate that this formulation of motor planning as a SOOC problem naturally accounts for optimal muscle co-contraction and impedance control without the need to estimate the state of the system during movement execution.

To introduce the SOOC theory, we first revisit the seminal work of [48]. Hogan considered the problem of maintaining the forearm in an unstable upright posture in presence of uncertainty and without feedback about the system state. The forearm was modeled as an inverted pendulum in the gravity field, actuated by a pair of antagonistic muscles as follows:

$$I\ddot{\theta} = T(u_1 - u_2) - K(u_1 + u_2)\theta - b\dot{\theta} + mgl_c \sin(\theta) + G\eta \tag{1}$$

where $\theta$ is the joint angle (0° being the upright orientation of the forearm and a dot above a variable standing for its time-derivative), $I$ is the moment of inertia, $m$, $l_c$, and $g$ are respectively the mass, length to the center-of-mass and gravity acceleration, $b$ is a damping parameter, and $\eta$ is some noise (typically Gaussian). Parameters $T$ and $K$ are constants –as well as the noise factor $G$ for the moment– and $u_i$ are the "neural" non-negative inputs to the flexor ($i = 1$) and extensor ($i = 2$) muscles. With this simplified model, Hogan showed that the optimal open-loop controls $(u_i(t)_{i=1..2})$ that should be used to maintain the forearm in the unstable upright position while minimizing an expected cost based on effort and variance led to some optimal amount of co-contraction (i.e. $u_1 = u_2 > 0$). The variable stiffness property of muscles, and the fact that stiffnesses of opposing muscles add, allowed to maintain this unstable posture even without on-line feedback about the actual system state. This minimal example captures a crucial feature for our subsequent theoretical developments: the controlled system to obtain this result involved interaction between control and state components (i.e. terms in $u_i \theta$). Without gravity ($g = 0$) or with linearization of gravitational torque (e.g. $\sin(\theta) \approx \theta$), this type of system is called "bilinear" in control theory and it will be the simplest class of systems for which the

SOOC framework makes original and relevant predictions regarding force and impedance planning. For linear control systems, which are often assumed in the motor control literature for simplicity, no difference with a deterministic approach would be observed. In the following, we build upon these ideas to model movement planning (not only posture as in Hogan's initial work) and extend the method to more general nonlinear dynamics (not only one degree-of-freedom or bilinear dynamics as in this example).

### Stochastic optimal open-loop control for bilinear systems

Consider stochastic dynamics with bilinear drift of the form:

$$d\mathbf{x}_t = [(A + \sum_{i=1}^{p} N_i u_i(t))\mathbf{x}_t + B\mathbf{u}(t)]dt + G(\mathbf{u}(t), t)d\omega_t \tag{2}$$

with $\omega_t$ being a m-dimensional standard Brownian motion. The stochastic state is denoted by $\mathbf{x}_t \in \mathbb{R}^n$ and the deterministic control is denoted by $\mathbf{u}(t) \in \mathbb{R}^p$. The matrix $G(\mathbf{u}(t), t) \in \mathbb{R}^{n \times m}$ can account for noise with both constant and signal-dependent variance.

In the simplest setting, our goal is to find the optimal open-loop control $\mathbf{u}(t)$ that minimizes a quadratic expected cost of the form:

$$C(\mathbf{u}) = \mathbb{E}[\int_0^{t_f} (\mathbf{u}(t)^T R\mathbf{u}(t) + \mathbf{x}_t^T Q\mathbf{x}_t)\, dt + \mathbf{x}_f^T Q_f \mathbf{x}_f]. \tag{3}$$

where $R$ is a positive definite matrix and $Q$, $Q_f$ are positive semi-definite matrices, all of appropriate dimensions. Note that because $\mathbf{u}(\cdot)$ is a deterministic function by hypothesis, the related integral value can be taken outside the expectation operator.

We assume that the system has an initial state distribution that is known, $\mathbf{x}_0$, at the initial time (hence state estimation from sensory feedback is at least required initially for motor planning). Time $t_f$ is the total movement duration, which can be fixed a priori or left free. For such a bilinear system, $\mathbf{x}_t$ will be a Gaussian process because the associated stochastic differential equation is actually linear (since $\mathbf{u}(t)$ is deterministic in the drift and diffusion). Therefore, the process $\mathbf{x}_t$ can be fully determined by propagation of its mean and covariance.

The propagation of the mean and covariance of the process $\mathbf{x}_t$ (denoted respectively by $\mathbf{m}(t) = \mathbb{E}[\mathbf{x}_t]$ and $P(t) = \mathbb{E}[\mathbf{e}_t \mathbf{e}_t^T]$ with $\mathbf{e}_t = \mathbf{x}_t - \mathbf{m}(t)$) are given by the following ordinary differential equations (see [50] for example):

$$\begin{cases} \dot{\mathbf{m}} &= (A + \sum_i N_i u_i)\mathbf{m} + B\mathbf{u} \\ \\ \dot{P} &= [A + \sum_i N_i u_i]P + P[A + \sum_i N_i u_i]^T + GG^T. \end{cases} \tag{4}$$

The latter equation shows explicitly the dependence of the covariance propagation on the control $\mathbf{u}$.

Next, a simple calculation shows that the expected cost $C(\mathbf{u})$ in Eq 3 can be rewritten as follows:

$$C(\mathbf{u}) = \int_0^{t_f} (\mathbf{u}^T R\mathbf{u} + \mathbf{m}^T Q\mathbf{m} + \text{trace}(QP))\, dt + \mathbf{m}_f^T Q\mathbf{m}_f + \text{trace}(Q_f P_f) \tag{5}$$

Therefore, we have just shown that the initial SOOC problem reduces to an exactly-equivalent DOC problem, the state of which is composed of the elements of the mean and covariance of the stochastic process $\mathbf{x}_t$.

It must be noted that this equivalent deterministic problem has nonlinear dynamics but a quadratic cost. This constraint of a quadratic cost is however not critical as any Lagrangian depending on the mean and covariance could be added to the original cost function. For example, the following more general type of costs could be considered as well:

$$C(\mathbf{u}) = \mathbb{E}[\int_0^{t_f} (L(\mathbf{m}(t), \mathbf{u}(t)) + \mathbf{x}_t^T Q \mathbf{x}_t) \, dt + \mathbf{x}_f^T Q_f \mathbf{x}_f]. \tag{6}$$

The deterministic term $L(\mathbf{m}, \mathbf{u})$ can be used for instance to implement a minimum hand jerk on the mean behaviour, in agreement with the deterministic control literature [5]. Note also that the term $\mathbf{x}_t^T Q \mathbf{x}_t$ could be replaced by $(\mathbf{x}_t - \mathbf{m}(t))^T \bar{Q} (\mathbf{x}_t - \mathbf{m}(t))$ to introduce a penalty on the state covariance alone in the equivalent DOC problem, in agreement with the minimum variance model [49]. Terminal state constraints or path constraints could also be added on the mean and covariance of the state process $\mathbf{x}_t$ without any difficulty but they are not described here. Typically, this could be useful to impose hard constraints on the mean state to reach and/ or on the covariance state to set a desired final accuracy. Such modeling choices will be illustrated in the subsequent arm movement and posture simulations, and can easily be handled within the equivalent DOC framework. Remarkably, optimal solutions of such DOC problems can be obtained via efficient existing numerical methods (e.g. [51]).

## Stochastic optimal open-loop control for general nonlinear systems

We now consider more general stochastic dynamics of the form

$$d\mathbf{x}_t = \mathbf{f}(\mathbf{x}_t, \mathbf{u}(t), t) \, dt + G(\mathbf{x}_t, \mathbf{u}(t), t) \, d\omega_t. \tag{7}$$

An example of such nonlinear system would be the system of Eq 1 (due to the gravitational torque). Multijoint arms also exhibit complex nonlinear dynamics due to inertial, centripetal, Coriolis, and gravitational torques. Nonlinearities will also arise for more advanced musculoskeletal models of the upper-limb. Therefore, we need a method to solve SOOC problems for the general class of nonlinear stochastic systems described in Eq 7.

Here we thus seek for a deterministic control $\mathbf{u}(t)$ minimizing the expectation of the above quadratic cost function (see Eq 6) and acting on the stochastic dynamics of Eq 7. The random process $\mathbf{x}_t$ is not necessarily Gaussian anymore. However, mean and covariance are still variables of major interest for movement control and their propagation would yield significant information about both mean behaviour and its variability. Actually we have shown in [47] that via statistical linearization techniques the control $\mathbf{u}(t)$ can be approximated by the solution of a DOC problem involving the propagation of the mean and covariance.

For instance, with Gaussian statistical linearization based on first order Taylor approximations, computations in [47] show that the dynamics of the mean and covariance in the corresponding DOC problem are:

$$\begin{cases} \dot{\mathbf{m}}(t) & = & \mathbf{f}(\mathbf{m}(t), \mathbf{u}(t), t), \\ \dot{P}(t) & = & \mathbf{F}(\mathbf{m}(t), \mathbf{u}(t), t) P(t) + P(t) \mathbf{F}(\mathbf{m}(t), \mathbf{u}(t), t)^T + \mathbb{E}[G(\mathbf{x}_t, \mathbf{u}(t), t) G(\mathbf{x}_t, \mathbf{u}(t), t)^T] \end{cases} \tag{8}$$

where $\mathbf{F}(\mathbf{m}(t), \mathbf{u}(t), t) = \frac{\partial \mathbf{f}}{\partial \mathbf{x}}(\mathbf{m}(t), \mathbf{u}(t), t)$.

If $G = G(\mathbf{u}(t), t)$ models constant and signal-dependent noise, then the latter expectation simply equates to $G(\mathbf{u}(t), t) G(\mathbf{u}(t), t)^T$. For a more general term, such as $G(\mathbf{x}_t, \mathbf{u}(t), t)$, more terms may be used to approximate the covariance propagation and the reader is referred to [52] for more information.

In summary, approximate solutions of the original SOOC problem can also be obtained from an associated DOC problem based on the mean and covariance of a process approximating the two first moments of the original process $\mathbf{x}_t$. Then, state-of-the-art DOC solvers can be used to find numerical solutions and model other constraints if desired (e.g. set a desired final positional variance or final mean position as a hard constraint. . .). The accuracy of these approximate solutions can be tested by simulating the original stochastic equation (Eq 7) with the obtained optimal control, and by comparing the evolution of the mean and covariance with Monte Carlo sampling techniques.

## Stochastic optimal open-loop control in the neural control of movement

The mathematical SOOC framework being formally introduced, we are now left with modeling choices to describe the effects of co-contraction. On the one hand, one may consider an end-effector or joint level description of the stiffness-like property of the neuromusculoskeletal system (e.g. [48] or the r- and c-commands in [14]). On the other hand, one may consider more advanced models representing the multiple muscles crossing each joint, the activation of which will modulate both the apparent stiffness of the musculoskeletal system and net joint torques (e.g. [53]). This choice is related to the hierarchical control hypothesis, as discussed in [4, 14] for instance. As this choice is still elusive, we will consider both joint and muscle levels of description. In particular, this will highlight the generality and consistency of the proposed theory beyond specific modeling choices.

**Joint level modeling: Explicit description of force and impedance planning.** In this paragraph, we extend Hogan's model presented above to account for the control of movement. Consider again the forearm model given in Eq 1. To simplify the derivations, we note that the state of a joint can be modified only in two ways: it can either (1) be moved to another position via changes of net torques or (2) be stiffened with no apparent motion via co-contraction [14]. Accordingly, the forearm dynamics can be rewritten as

$$I\ddot{\theta} = \tau(t) - K_s\kappa(t)(\theta(t) - \Theta(t)) - K_d\sqrt{\kappa(t)}(\dot{\theta}(t) - \dot{\Theta}(t)) - b\dot{\theta} + mgl_c \sin(\theta) + G\eta, \quad (9)$$

where $\tau(t) \in \mathbb{R}$ is the net joint torque and $\kappa(t) \in \mathbb{R}_+$ modulates the joint stiffness (two control variables). The function $\Theta(t)$ serves as a reference trajectory which is not present in Hogan's original formulation but is critical to change the limb's working position. A potential justification could be that intact muscles behave like "nonlinear springs with modifiable zero-length" [54]. Hence, we assume that the resultant joint-level effect of this characteristics allows to set an equilibrium joint position or trajectory. In addition to stiffness, damping also seems to be modified through co-contraction [55]. We thus added a term in factor of $K_d = \sqrt{IK_s}$ such that the damping ratio, in terms of a second-order model, was constant (here equal to 1/2, e.g. see [56, 57]).

In order to determine the reference trajectory $\Theta(t)$, there are several choices. For instance, one could consider a third control variable to choose $\Theta(t)$, by adding an equation such as $\dot{\Theta} = u(t)$. This reference trajectory might be very simple (e.g. steady state or linear). The drawback is to introduce a third control variable, the choice of which seems rather elusive (e.g. what cost function on it). Alternatively, a better choice may be to consider reference trajectories that are themselves solutions of the joint-level dynamics. In this case, we assume that $\Theta(t)$ satisfies the rigid body dynamics

$$I\ddot{\Theta} = \tau(t) - b\dot{\Theta} + mgl_c \sin(\Theta) \quad (10)$$

with $\Theta(0) = \theta(0)$. Hence, if we define $\Delta(t) = \theta(t) - \Theta(t)$, one can derive the following system

from Eqs 9 and 10:

$$
\begin{cases}
I\ddot{\Theta} &= \tau(t) - b\dot{\Theta} + mgl_c \sin(\Theta) \\
I\ddot{\Delta} &= -K_s\kappa(t)\Delta - K_d\sqrt{\kappa(t)}\dot{\Delta} - b\dot{\Delta} + mgl_c(\sin(\Delta + \Theta) - \sin(\Theta)) + G\eta
\end{cases}
\tag{11}
$$

with $\Delta(0) = 0$, $\Theta(0) = \theta(0)$ and $\dot{\Delta}(0) = 0$, $\dot{\Theta}(0) = 0$. The advantage of this formulation, relying on a reference angle $\Theta$ and deviations $\Delta$ from it, is that only two controls are needed, namely $\tau(t)$ that specifies the net joint torque and $\kappa(t)$ that specifies the joint stiffness (and damping). As such, this modeling implements a separate control of force (via reciprocal commands) and impedance (via co-contraction commands), which is compatible with several experimental findings [18, 19, 58, 59].

Now assume that the goal is to minimize an expected cost of the form

$$
C(\tau, \kappa) = \mathbb{E}\left[\int_0^{t_f}(\tau^2 + \alpha\kappa^2 + q_{var}(\Delta^2 + q_v\dot{\Delta}^2))dt + q_{var}(\Delta_f^2 + q_v\dot{\Delta}_f^2)\right].
\tag{12}
$$

where the cost elements in $\Delta$ and $\dot{\Delta}$ penalize deviations from the reference trajectory (variance), and the control costs penalize effort. Weight factors $\alpha$, $q_v$ and $q_{var}$ can be chosen to adjust the optimal behavior of the system. Typically, optimal solutions will yield minimal net joint torque and impedance to remain close to the reference trajectory to some extent determined by the weight $q_{var}$ (and $q_v$).

In the present form, the dynamics of $\Theta$ and $\Delta$ are coupled by the gravitational term. To derive an interesting result, let us focus on horizontal movements for a moment. The system then simplifies to:

$$
\begin{cases}
I\ddot{\Theta} &= \tau(t) - b\dot{\Theta} \\
I\ddot{\Delta} &= -K_s\kappa(t)\Delta - K_d\sqrt{\kappa(t)}\dot{\Delta} - b\dot{\Delta} + G\eta.
\end{cases}
\tag{13}
$$

This system is actually a controlled SDE with a 4-D state vector $(\Theta, \dot{\Theta}, \Delta, \dot{\Delta})^\top$. Interestingly, the first two states are deterministic (in $\Theta$) and noise only affects deviations from these reference states (in $\Delta$, which we now rewrite $\Delta_t$ to stress that it is a random variable). Remarkably, the original SOOC problem is completely decoupled in this case. On the one hand, we have a deterministic sub-problem with dynamics in $\Theta$ and cost in $\tau^2$. On the other hand, we have a stochastic sub-problem with dynamics in $\Delta_t$ and cost in $\kappa^2$ and $\Delta_t, \dot{\Delta}_t$. This SOOC problem only involving the mean and covariance of the state $(\Delta_t, \dot{\Delta}_t)$ can be solved by deriving an equivalent DOC problem as described before. Since the mean of $(\Delta_t, \dot{\Delta}_t)$ is zero given the initial conditions, propagation of the mean can be neglected for this part. Regarding $\Theta(t)$, since it is a deterministic variable, the propagation of its covariance can be neglected. In summary, the components $\Theta(t), \dot{\Theta}(t)$ correspond to the mean of the actual state $\theta_t, \dot{\theta}_t$, and the covariance of $\Delta_t, \dot{\Delta}_t$ corresponds to the covariance of the actual state $\theta_t, \dot{\theta}_t$. Therefore, the net torque $\tau(t)$ controls the mean of the stochastic process in $\theta_t, \dot{\theta}_t$ whereas $\kappa(t)$ controls its covariance independently. As the process in Eq 13 is Gaussian, it is thus fully characterized and controlled.

These derivations are useful to understand the functioning of the model but, in more realistic scenarios, the above decoupling will not hold anymore. This is in no way an issue or limitation because more general cases can be easily handled within the SOOC framework. For instance, if gravity is not neglected, Eq 11 can be used together with Eq 12 to formulate a

nonlinear SOOC problem that can be resolved with the methods described in the previous subsections.

Finally, these considerations may be reminiscent of the equilibrium point theory [60]. However, the approach differs from equilibrium point theory in the sense that a feedforward torque controller $\tau(t)$ is assumed (hence our approach requires "internal models"). It nevertheless fits with some aspects of the equilibrium point theory in the sense that a "virtual" reference trajectory $\Theta$ is planned together with time-varying impedance parameters (which may be tuned in practice via co-contraction of opposing muscles). As such, $\tau(t)$ and $\kappa(t)$ might resemble the c- and r-commands described in [14] even though movement is not generated only by the viscoelastic properties of the musculoskeletal systems and shifts in equilibrium points/trajectories in our case (see also Discussion).

**Muscle level modeling: Implicit description of force and impedance planning.** Here we use more advanced models of the musculoskeletal system. In this work, we used the muscle model proposed by Katayama and Kawato [53] and assume that a feedforward motor command can be sent to each muscle individually.

**Single-joint arm.** For a single-joint system like the forearm moving in the horizontal plane, Katayama and Kawato's model writes as follows:

$$I\ddot{\theta} \quad = \quad \tau_1 + \tau_2 + G\eta \tag{14}$$

where $\tau_1$ and $\tau_2$ are the muscle torques that are respectively functions of muscle activations $u_1$ and $u_2$, defined as follows:

$$
\begin{aligned}
\tau_i \quad &= \quad -a_i T_i, \ \ i = 1 \text{ or } 2 \\
T_i \quad &= \quad (k_0 + k_i u_i)(r_i u_i + l_0 - l_{m_i} + a_i \theta) + (b_0 + b_i u_i) a_i \dot{\theta}, \ \ i = 1 \text{ or } 2
\end{aligned}
\tag{15}
$$

In this case, the system state is $\mathbf{x} = (\theta, \dot{\theta})^\top$. Muscle parameters were taken from [53]. Here we have $I = 0.0588$ kg.m$^2$, $k_i = 1621.6$ N/m, $k_0 = 810.8$ N/m, $b_i = 108.1$ N.s/m, $b_0 = 54.1$ N.s/m, $a_i = 2.5$ cm, $r_i = (-1)^i \times 2.182$ cm for i = 1..2, $l_{m_1} - l_0 = 5.67$ cm and $l_{m_2} - l_0 = 0.436$ cm. We also set $m = 1.44$ kg, $l_c = 0.21$ m, and the forearm length was 0.35 m.

In this model, the muscle torque thus depends on the muscle activation as well as on the angular position and velocity. The muscle activations can therefore modulate both the net joint torque and the joint viscoelasticity, in particular via muscle co-contraction.

**Two-joint arm.** A two degrees-of-freedom (dof) version of the arm with 6 muscles was also considered to simulate planar arm reaching movements, corresponding to the full model of [53]. The state of the arm is then $\mathbf{x}^\top = (\mathbf{q}^\top, \dot{\mathbf{q}}^\top) \in \mathbb{R}^4$ where $\mathbf{q} = (\theta_1, \theta_2)^\top$ denotes the joint angle vector (1st component for shoulder and 2nd component for elbow) and $\dot{\mathbf{q}} = (\dot{\theta}_1, \dot{\theta}_2)^\top$ denote the corresponding joint velocity vector. The dynamics of the arm follows a rigid body equation of the form:

$$\ddot{\mathbf{q}} = \mathcal{M}^{-1}(\mathbf{q})(\tau(\mathbf{q}, \dot{\mathbf{q}}, \mathbf{u}) - \mathcal{C}(\mathbf{q}, \dot{\mathbf{q}})\dot{\mathbf{q}}) \tag{16}$$

where $\mathcal{M}(\mathbf{q})$ is the inertia matrix, $\mathcal{C}\dot{\mathbf{q}}$ is the Coriolis/centripetal term, $\tau$ is the net joint torque vector produced by muscles and $\mathbf{u} \in \mathbb{R}^6$ is the muscle activation vector (restricted to be open-loop/deterministic in this work).

The net joint torque vector is a function $\tau(\mathbf{q}, \dot{\mathbf{q}}, \mathbf{u})$ depending on the moment arms (assumed constant) and on the muscle lengths/velocities expressed as affine functions of the joint positions and velocities as in Eq 15. All the parameters of the complete model with 6 muscles can be found in the Tables 1, 2, and 3 in [53]. Finally, by introducing noise ($\boldsymbol{\omega}_t$ is a 2-dimensional standard Brownian motion), we obtain the following SDE modeling the noisy

musculoskeletal dynamics of a multijoint arm:

$$d\mathbf{x}_t = \mathbf{f}(\mathbf{x}_t, \mathbf{u}(t))dt + G d\omega_t \qquad (17)$$

with

$$\mathbf{f}(\mathbf{x}_t, \mathbf{u}(t)) = \begin{pmatrix} \dot{\mathbf{q}}_t \\ \mathcal{M}^{-1}(\mathbf{q}_t)(\tau(\mathbf{q}_t, \dot{\mathbf{q}}_t, \mathbf{u}(t)) - \mathcal{C}(\mathbf{q}_t, \dot{\mathbf{q}}_t)\dot{\mathbf{q}}_t) \end{pmatrix} \qquad (18)$$

and $G$ is a matrix allowing to define the coupling of the system to the noise.

## Results

In this section, we consider simulations accounting for results of several experimental findings about the planning of force and impedance as well as on the role of muscle co-contraction in posture and movement control. Different models, types of noise, and cost functions will be used to illustrate the flexibility of the framework in making consistent predictions about the role of co-contraction and impedance for the open-loop control of stochastic systems.

### Co-contraction planning in 1-dof motor tasks

**Unstable postural task with the forearm.** In Hogan's study described above [48], the maintenance of a human forearm in an upright posture was considered. Hogan described a system controlled by a pair of agonist/antagonist muscles and showed that co-contraction was a strategy used by participants to maintain such an unstable posture. Here, we reconsider this task to test our framework with this simple starting example. As already mentioned, Hogan modeled the forearm as an inverted pendulum in the gravity field driven by opposing muscles having the essential force-length dependence of real muscles (see Eq 1). Here we considered two scenarios tested in Hogan's experiment depending on whether the forearm was loaded ($m_{load}$ = 2.268 kg attached at the hand) or unloaded. To efficiently resolve the problem, we showed in [47] that via statistical linearization techniques we can get a bilinear system as in Eq 2 (which amounts to linearize the gravitational torque, i.e. $\sin\theta \approx \theta$ with a small angle hypothesis), with matrices defined as follows:

$$A = \begin{pmatrix} 0 & 1 \\ \dfrac{mgl_c}{I} & -\dfrac{b}{I} \end{pmatrix}, \; B = \begin{pmatrix} 0 & 0 \\ \dfrac{T}{I} & -\dfrac{T}{I} \end{pmatrix}, \; \text{and} \; N_i = \begin{pmatrix} 0 & 0 \\ -\dfrac{K}{I} & 0 \end{pmatrix}, \; i = 1 \text{ or } 2. \qquad (19)$$

where the parameters are defined by Eq 1.

In our simulations, noise was taken of constant variance and acting at acceleration level, $G = (0, .1)^\top$. We set $T = K = 1$ and considered a fixed damping parameter $b = 1$ Nms/rad. The cost was defined as $R = \text{diag}(1, 1)$, $Q = Q_f = \text{diag}(10^4, 10^3)$ (see Eq 3). These weights were chosen such as to make effort and variance costs of comparable magnitude (note that angles are in radians). We assumed that the system was at state $\mathbf{x}_0 = \mathbf{0}$ at initial time $t = 0$ with zero covariance. The goal of the task was to maintain the inverted pendulum around this position without on-line sensory feedback for $t_f = 5$ s while minimizing a compromise between variance (cost depending on $Q$, $Q_f$) and effort (cost depending on $R$). In the equivalent DOC problem, the final covariance $P_f$ was left free whereas the final mean state was set to zero ($\mathbf{m}_f = 0$). Results of simulations are reported in Fig 1. In these graphs, one can see that an optimal level of stiffness is achieved to stabilize the forearm in this unstable posture. We checked that, if there was no co-contraction, the forearm would fall in about one second due to the combined effects of noise and gravity (remind that feedback control is prevented). Therefore, co-contraction

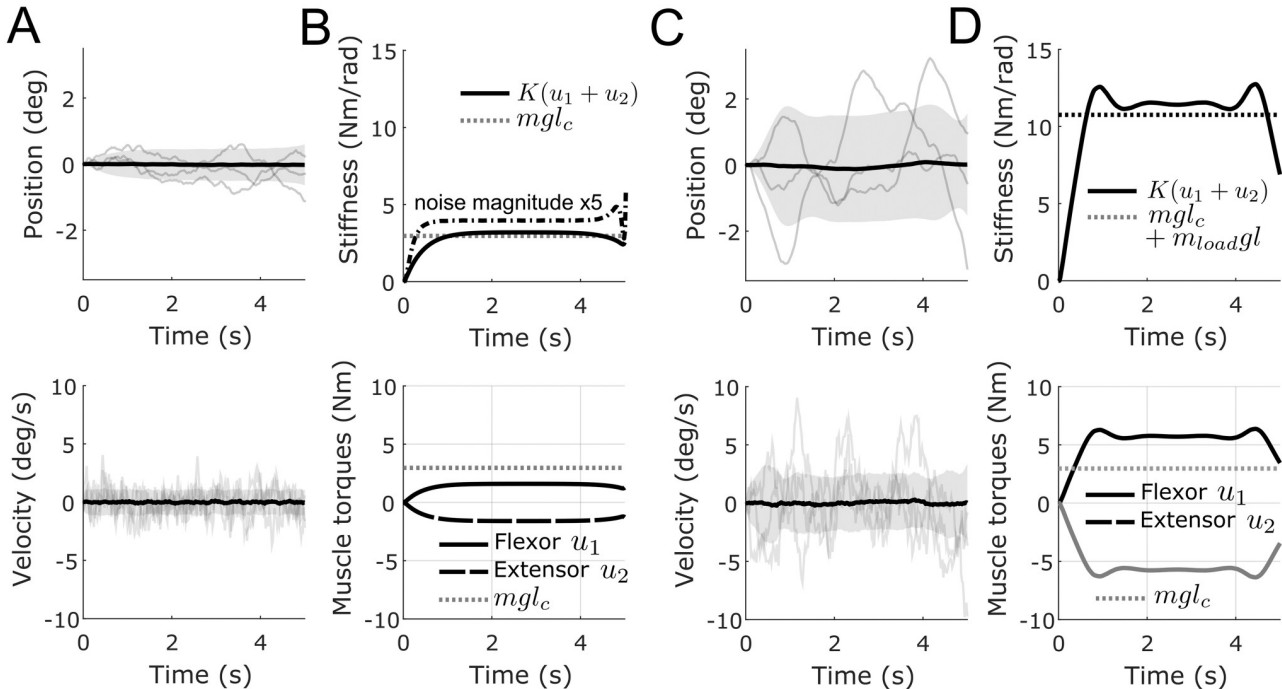

**Fig 1. Co-contraction during maintenance of an upright forearm posture.** A. Position (in degrees) and velocity (in degrees per second) for the unloaded case. Thick black lines depict the means and shaded areas depict standard deviations (from 500 samples). Three single trajectories are displayed to illustrate their stochastic nature. B. Corresponding optimal joint stiffness (solid black line) and flexor/extensor muscle torques (solid and dashed respectively) for the unloaded case. In dotted gray line, the "divergent" stiffness level that co-contraction must overcome for stability is depicted (i.e. $mgl_c$ in our case). In dashed-dotted black line, we also report the optimal stiffness when noise magnitude is increased by a factor 5, i.e. $G = (0, .5)^\top$. C-D. Same information for the loaded case where $I$ was increased by $m_{load} l^2$ and $mgl_c$ was increased by $m_{load} gl$. A significant increase of stiffness, originating from a larger co-contraction of flexor and extensor muscles, can be noticed. Parameters: $m = 1.44$ kg, $m_{load} = 2.268$ kg, $l = 0.35$ m, $l_c = 0.21$ m, $I = 0.0588$ m.kg$^2$ and $g = 9.81$ m/s$^2$.

creates a resultant stiffness that is just enough to compensate the task instability. Note that if noise magnitude is larger, a larger co-contraction becomes optimal (dashed-dotted line in Fig 1B). This change in co-contraction with noise magnitude agrees with a study of Hasson [61]. In the loaded case, the destabilizing gravitational torque increases and the optimal co-contraction level becomes larger to counteract it. Remarkably, it can be observed that, like in Hogan's original work [48], the activation of the flexor muscle in the loaded case is larger than the corresponding activation of that muscle that would be necessary to maintain the forearm horizontal in the unloaded case (dotted line in the bottom-right panel). Note that the exact shape of the optimal solutions depends on the terminal state constraints (on mean and covariance), the weights in the cost function (i.e. the effort/variance trade-off) and the level of noise. The lack of steady-state behavior near the end of the simulation is in particular due to the finite time-horizon used in these simulations and the associated terminal cost/constraints.

## Reaching task with the forearm

**Joint-level modeling.** Here we consider single-joint pointing movements performed with the forearm. We first use the joint-level description of force and impedance derived in Eqs 9–13. For these simulations, we focus more specifically on the controlled system described in Eq 13 and on the experimental data reported in [57, 62]. Bennett's main findings were that the elbow joint stiffness varies within point-to-point movements (either cyclical or discrete). The forearm was found to be overall quite compliant (measured stiffness ranging between 2 Nm/

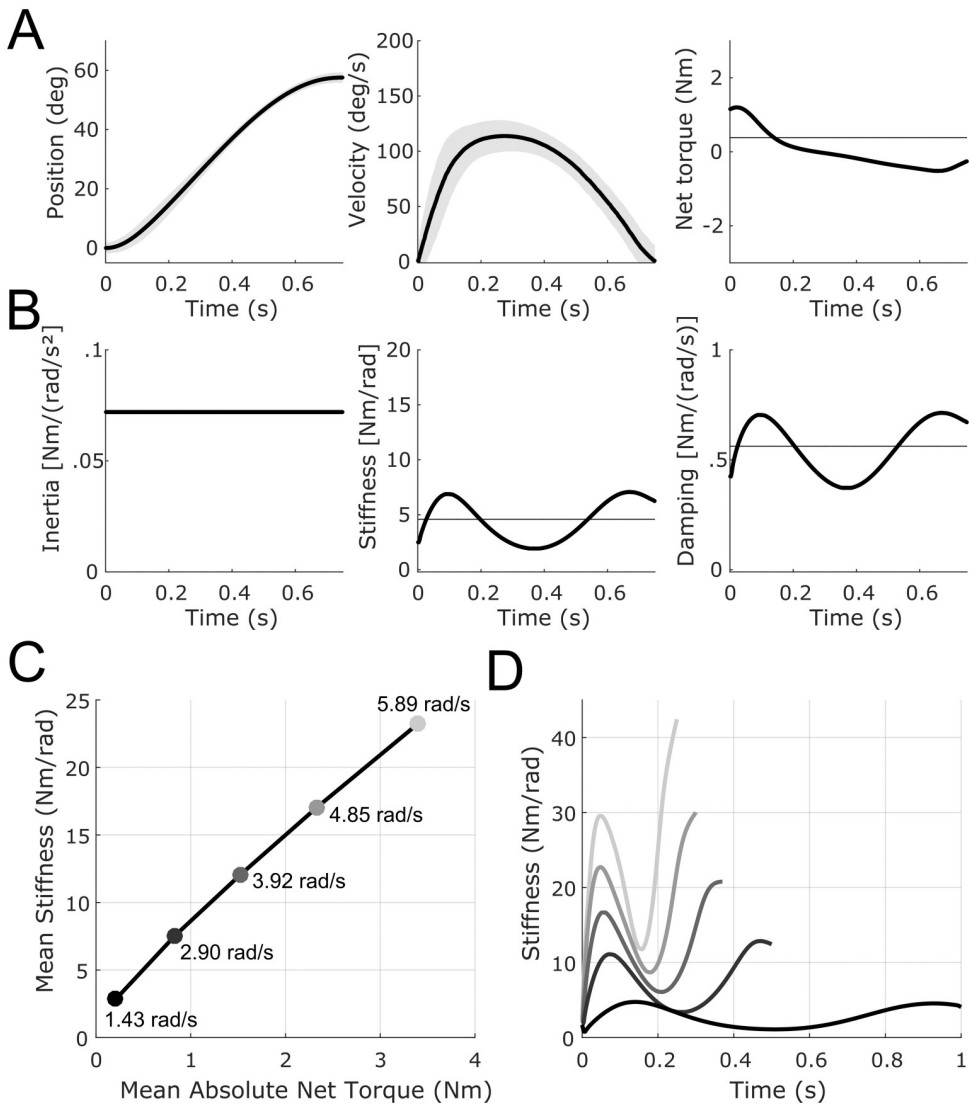

**Fig 2. Simulations of a pointing movement with the forearm (elbow extension of 1 rad).** A. Optimal trajectories. Angular displacement and velocity profiles (mean in thick lines and standard deviation in shaded areas) and net torque responsible of the joint motion (third column). B. Optimal impedance. Inertia (constant for this single-joint system), stiffness and damping are depicted. Time-varying joint stiffness and damping, part of the optimal open-loop motor plan in our model, are responsible of robustness of motor behavior around the mean behavior (without needing on-line feedback for that purpose). Note an increase of stiffness at the end of the motion, to improve accuracy on target, in agreement with experimental data. Values can be quantitatively compared to [62]. Time-average values are represented by horizontal lines. C. Relationship between time-average net torques and time-average stiffnesses for movements of different speeds. Peak velocity is indicated in rad/s. An approximately linear relationship is observed as in [57]. D. Corresponding time-varying stiffness profiles for the different movement speeds. In panels A and B, parameters were as follows: $q_{var} = 10^4$, $\alpha = 1$, $q_v = 0.01$, $t_f = 0.75$, and $I = 0.072$. In panels C and D, same parameters but with $t_f$ ranging between 0.25 s and 1.0 s to generate movements of different speeds.

rad and 20 Nm/rad). Yet, stiffness significantly increased when the hand approached the target and stiffness had minimal values near peak velocity. Additionally, mean joint stiffness was found to increase with peak velocity and to increase almost linearly with the magnitude of net joint torque. In Fig 2 we replicated these main observations within our framework. Fig 2A and 2B shows the optimal behavior for movements executed in presence of signal-dependent noise

(proportional to net torque $\tau$). Because the task in [62] involved cyclical forearm movements, we imposed equal initial and final covariances. We also chose $b = 0$ since damping was already modulated together with stiffness in this model such as to get a constant damping ratio as suggested in [57] (damping however seemed harder to identify in general but it tended to fluctuate around 0.5 Nms/rad in experimental data). We penalized the integral of the square of the controls (effort) plus a covariance term involving position and velocity in the cost function (see Eq 12). We considered multiplicative noise acting in torque space in these simulations (20% of net torques, i.e. $G = G(\tau) = \frac{1}{I}\left(0, 0, 0, 0.2\tau(t)\right)^{\top}$).

In Fig 2C and 2D, we varied movement duration to test the effect of speed on joint stiffness. Results can be compared to [57, 62, 63]. We found that, indeed, stiffness tends to increase almost linearly with net torque (which also increases with speed). These values have been compared to phasic and tonic EMG data in experimental works. However, this joint-level description of force and impedance planning does not allow to see the origin of stiffness in terms of muscle co-contraction. Therefore, we next consider muscle models to further investigate the co-contraction phenomenon in reaching arm movements.

**Muscle-level modeling.** Here we consider the musculoskeletal arm model proposed in [53] (Eq 14) to simulate elbow flexion movements in the horizontal plane (hence with $g = 0$). This is a muscle-level description of force and impedance planning. We focus on the experimental results of [64] which showed that subjects can reach faster while preserving accuracy if asked to co-contract opposing muscles. A signal-dependent noise model was defined here as in [29] in order to model that co-contraction does not lead to increased variability [16] as it would be the case for a standard signal-dependent noise model. The noise model was as follows:

$$G(\mathbf{u}(t)) = \begin{pmatrix} 0 & d(|u_1(t) - u_2(t)|^{1.5} + 0.01|u_1(t) + u_2(t)|^{1.5}) \end{pmatrix}^{\top} \tag{20}$$

where $d$ is a factor to set the overall magnitude of this signal-dependent noise (here we fixed $d = 4$ in simulations because it yielded good quantitative predictions of the empirical variability found in such fast reaching movements).

The cost function of the associated deterministic problem (Eq 5) was as follows:

$$C = \int_0^{t_f} \mathbf{u}(t)^{\top}\mathbf{u}(t)\,\mathrm{dt} + \mathrm{trace}(Q_f P_f) \tag{21}$$

with $Q_f = q_{var}\mathrm{diag}(1, 0.1)$. The term $Q_f$ simply penalizes the terminal state covariance and $q_{var}$ is a free parameter. We further set $\mathbf{m}_0 = (25°,0)^T$ and $\mathbf{m}_{targ} = (65°,0)^T$ as the initial and final mean positions of the reaching task as in [64]. The initial covariance $P_0$ was zero and the final covariance $P_f$ was left free but minimized because the goal of the task was to reach a target of width 5° (given that the amplitude of the movement was 40°, the index of difficulty was 4 bits for this task). As such, this cost function implements an effort-variance compromise as suggested in [64, 65].

Fig 3 shows the results of simulated pointing movements. In Fig 3A, we set $t_f = 475$ ms as in Missenard's experiment and $q_{var} = 50$. With these settings we reproduced quite well the spontaneous behavior of subjects in this experiment, which is representative of what occurs in a standard Fitts' like paradigm. For instance, peak velocities (PV) were about 140 deg/s and the index of co-contraction (IC) was about 20% in experimental data (index of co-contraction was defined as in [59, 64]). In our simulations we obtained PV of 139.75 deg/s and IC of 22.35% (see Table 1). Yet, when asked to co-contract to an IC of ~80%, subjects in [64] were able in practice to perform the task with greater speeds without apparent loss of accuracy. We replicated this condition in Fig 3B by setting $t_f = 400$ ms as in Missenard's experiment and

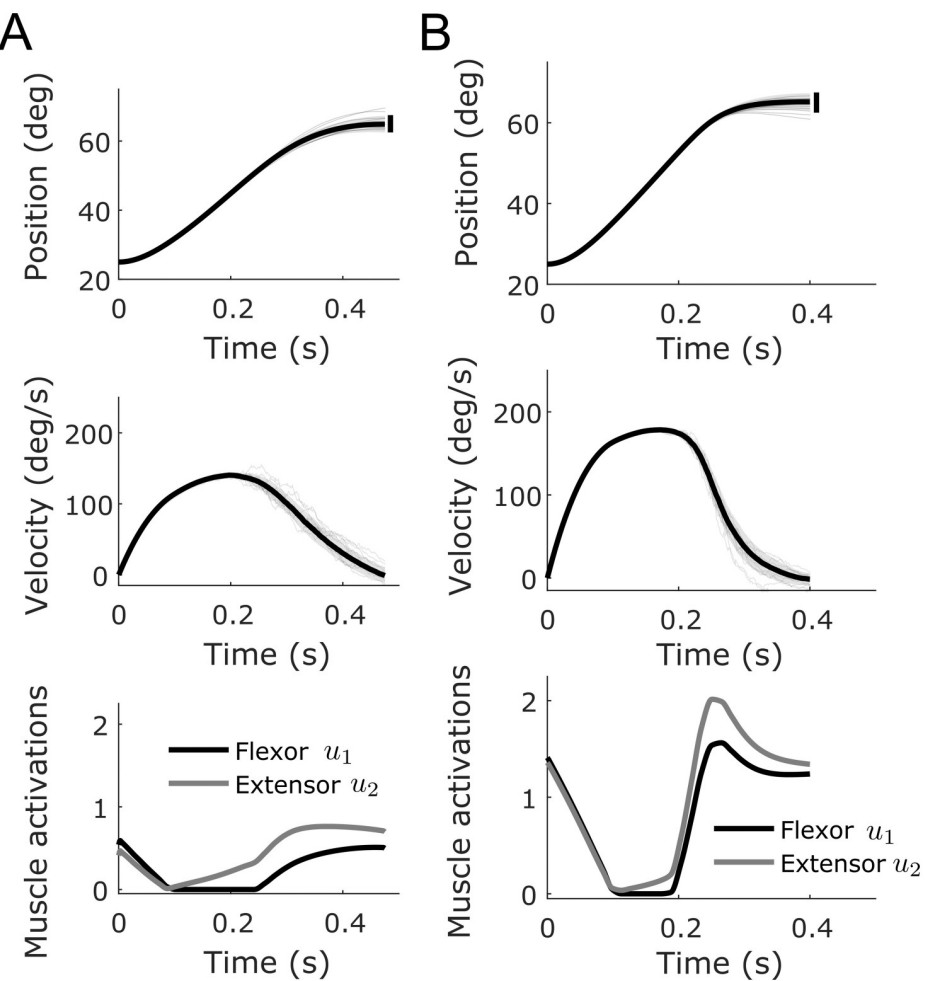

**Fig 3. Simulations of pointing movements following the experiment of [64].** A. Simulation with $t_f$ = 475 ms and $q_{var}$ = 50. The mean position and velocity (thick black traces) are reported. Shaded areas correspond to standard deviations. Some individual trajectories are also depicted with thin gray traces. The corresponding optimal open-loop muscle activations are reported (black for flexor and gray for extensor). Co-contraction is necessary to achieve the requested accuracy as actually observed in Fitts-like reaching experiments. B. Simulation with $t_f$ = 400 ms and $q_{var}$ = 500. Same information than in panel A. It is seen that with even more co-contraction, an acceptable accuracy can also be achieved but for faster movements. Note the asymmetry of velocity profiles, with a longer deceleration than acceleration, which is also typical of Fitts' instructions [66].

**Table 1. Parameters corresponding to the simulated movements of Fig 3.** In all cases, initial and final positions were 25˚ and 65˚ (amplitude of 40˚). The target width was 5˚ such that the index of difficulty (ID) was 4 bits. The reference end-point standard deviation can thus be 2.5˚. The effort is measured as the quadratic cost in **u** according to Eq 21.

| $t_f$ (ms) | EPstd (deg) | PV (deg/s) | $q_{var}$ | Effort ($\times 10^{-2}$) | IC (%) |
|---|---|---|---|---|---|
| 475 | 2.51 | 139.75 | 50 | 16.99 | 22.35 |
| 400 | 2.41 | 161.56 | 50 | 27.36 | 27.36 |
| 400 | 1.89 | 178.87 | 500 | 94.16 | 80.15 |
| 475 | 2.82 | 134.79 | 1 | 4.03 | <0.1 |
| 400 | 3.35 | 146.47 | 1 | 5.60 | <0.1 |

$q_{var}$ = 500. Because of signal-dependent noise, going faster increases noise magnitude and endpoint variance unless co-contraction is used. With these parameters, we obtained a PV of 178.87 deg/s (compared to about 180 deg/s in [64]) and an IC of 80.15%. Even though movements were faster, the positional standard deviation of the endpoint (EPstd) was 1.89 deg – hence smaller than half the target's width such that the task could be achieved successfully on successive trials–. Therefore, there should be no more overshoots or undershoots in this condition, as was observed in [64]. Table 1 shows that this improvement of speed at comparable accuracy is highly costly due to a clear co-contraction of agonist/antagonist muscles (see effort column), especially at the end of the movement (but co-contraction also appears at its beginning). For comparison, for strategies without co-contraction (e.g. obtained by setting a small weight, e.g. $q_{var}$ = 1), positional standard deviations of the end-point would be respectively 2.82 deg and 3.35 deg for movements times of 475 ms and 400 ms (see Table 1). This justifies why a minimal level of co-contraction is indeed required to perform the task accurately enough (again, with our open-loop control assumption). The fact that a significant co-contraction appears at the beginning and at the end of the movement agrees well with the literature [17, 62]. This example confirms that a trade-off between effort, speed, and accuracy may be prevalent in Fitts-like tasks, i.e. when subjects are instructed to perform the task as fast and as accurately as possible.

## Co-contraction planning in 2-dof motor tasks

Here we consider 2-dof arm reaching tasks and the musculoskeletal model described in [53] and Eqs 16–18. This model contains 4 single-joint muscles acting around the shoulder and elbow joints and 2 double-joint muscles. It has been shown to capture the basic stiffness properties of the human arm and has been investigated to evaluate the equilibrium point hypothesis originally. Here we used this model to test our SOOC framework with a quite advanced musculoskeletal model and to see whether co-contraction may be an optimal strategy to regulate the limb's mechanical impedance in open-loop in certain tasks.

**Two-link arm reaching task in a divergent force field.**   Burdet and colleagues found that subjects succeeded in performing accurate pointing movements in an unstable environment by selective muscle co-contraction [16, 18]. In their experiment, participants had to point to a target placed in front of them with a force field applying perturbations proportional to their lateral deviation from a straight line. Because the hand would start with random lateral deviation due to motor noise, it was not possible for the subjects to predict whether the arm would be pushed to the left or to the right during movement execution. The strength of the perturbation force (proportional to the extent of lateral deviation) and delays in sensorimotor loops would prevent participants from using an optimal closed-loop control strategy that requires accurate estimation of the system state to function (e.g. [67]). Instead, experimental data clearly showed that the solution of the participants was to stiffen their limb, in particular via co-contraction mechanisms and in a feedforward manner (e.g. participants kept co-contracting when the divergent force field was unexpectedly removed) [18, 35, 68].

Here we used the Eqs 17 and 18 to model the arm dynamics but we added the external perturbation force applied to the endpoint and had to consider an appropriate cost function to model the task. More precisely, a term

$$\ddot{\mathbf{q}}_{ext} = \mathcal{M}^{-1}(\mathbf{q}) J(\mathbf{q})^{\top} \begin{pmatrix} F_{ext} \\ 0 \end{pmatrix} \qquad (22)$$

was added to the right-hand side of Eq 16, with $J$ being the Jacobian matrix of the arm and

$F_{ext} = \beta x$ being the external force ($x$ is the Cartesian position of the hand along the horizontal axis and $\beta = 40\ Nm^{-1}$ in our simulations).

The cost function was designed as

$$C(\mathbf{u}) = \mathbb{E}\big[\int_0^{t_f} L(\mathbf{m}, \mathbf{u})\,dt + q_{var}\phi(\mathbf{x}_f)\big] \tag{23}$$

where $L(\mathbf{m}, \mathbf{u}) = \mathbf{u}^\top\mathbf{u} + \frac{1}{2}(\ddot{x}^2 + \ddot{y}^2)$, $\ddot{x}$ and $\ddot{y}$ being the mean Cartesian accelerations of the endpoint along the $x$ and $y$ axes respectively (i.e. functions of $\mathbf{m}$ and $\mathbf{u}$, which can be easily computed from the forward kinematic function), and $\phi(\mathbf{x}_f)$ is a function penalizing the covariance of the final state $\mathbf{x}_f$ ($q_{var}$ is a weighting factor). Hence this cost is a trade-off between minimum effort/variance and maximum smoothness in Cartesian space (e.g. see [5, 7, 69]). In these simulations, the smoothness term was needed because the minimum effort solution for this musculoskeletal model led to hand trajectories that were too curved compared to normal human behavior in the task (even without force field). For the variance term, we penalized the final positional variance in task space by defining

$$\phi(\mathbf{x}_f) = [J(\mathbf{m}_{q,f})(\mathbf{q}_f - \mathbf{m}_{q,f})][J(\mathbf{m}_{q,f})(\mathbf{q}_f - \mathbf{m}_{q,f})]^\top \tag{24}$$

where $\mathbf{m}_{q,f}$ is the mean of the final position of the process and $\mathbf{q}_f$ is a 2-D random vector composed of final joint positions extracted from $\mathbf{x}_f$.

The expectation of $\phi(\mathbf{x}_f)$ can be rewritten as a function of the final mean $\mathbf{m}_f$ and covariance $P_f$ as $\Phi(\mathbf{m}_f, P_f) = J(\mathbf{m}_{q,f})P_{\mathbf{q},f}J(\mathbf{m}_{q,f})^\top$ where $P_{\mathbf{q},f}$ denotes the 2x2 covariance matrix of joint positions. Finally, the expected cost function can be rewritten as follows:

$$C(\mathbf{u}) = \int_0^{t_f} L(\mathbf{m}, \mathbf{u})\,dt + q_{var}\Phi(\mathbf{m}_f, P_f). \tag{25}$$

The latter cost was used in the deterministic optimal control problem that approximates the solution to the original SOOC problem.

A simple noise model was considered in these simulations:

$$G(\mathbf{x}_t) = \begin{pmatrix} \mathrm{diag}(0,0) \\ \mathcal{M}^{-1}(\mathbf{q}_t)\mathrm{diag}(\sigma_1, \sigma_2) \end{pmatrix}, \tag{26}$$

where the parameters $\sigma_1$ and $\sigma_2$ were used to set the magnitude of constant additive noise (which was assumed to act in torque space, hence the inverse of the inertia matrix in the expression of $G$).

Results of simulations are reported in Fig 4 and Table 2. In these simulations, we set $t_f = 0.75$ s according to the data of [16]. The initial arm's position was located at (0,0.30) in Cartesian coordinates and the target was at (0,0.55). Noise magnitude was set to $\sigma_1 = \sigma_2 = 0.025$ in Eq 26.

Overall we found that it was possible to perform this unstable reaching task without on-line estimation of the actual system state by co-contracting pairs of opposing muscles (see Fig 4B). This finding agrees with [18, 68]. Muscle co-contraction increased when the endpoint variance was penalized more strongly (but at the cost of a greater effort) and when the divergent force field had a greater magnitude (see mean endpoint stiffness along the x-axis in Table 2). Co-contraction also increased when noise magnitude was increased, all other parameters being equal (Table 2). As a rule of thumb, endpoint stiffness was found to increase with (1) the magnitude of the divergent force field, (2) the weight of the variance cost and (3) the magnitude of noise, in agreement with experimental observations [17, 61, 70].

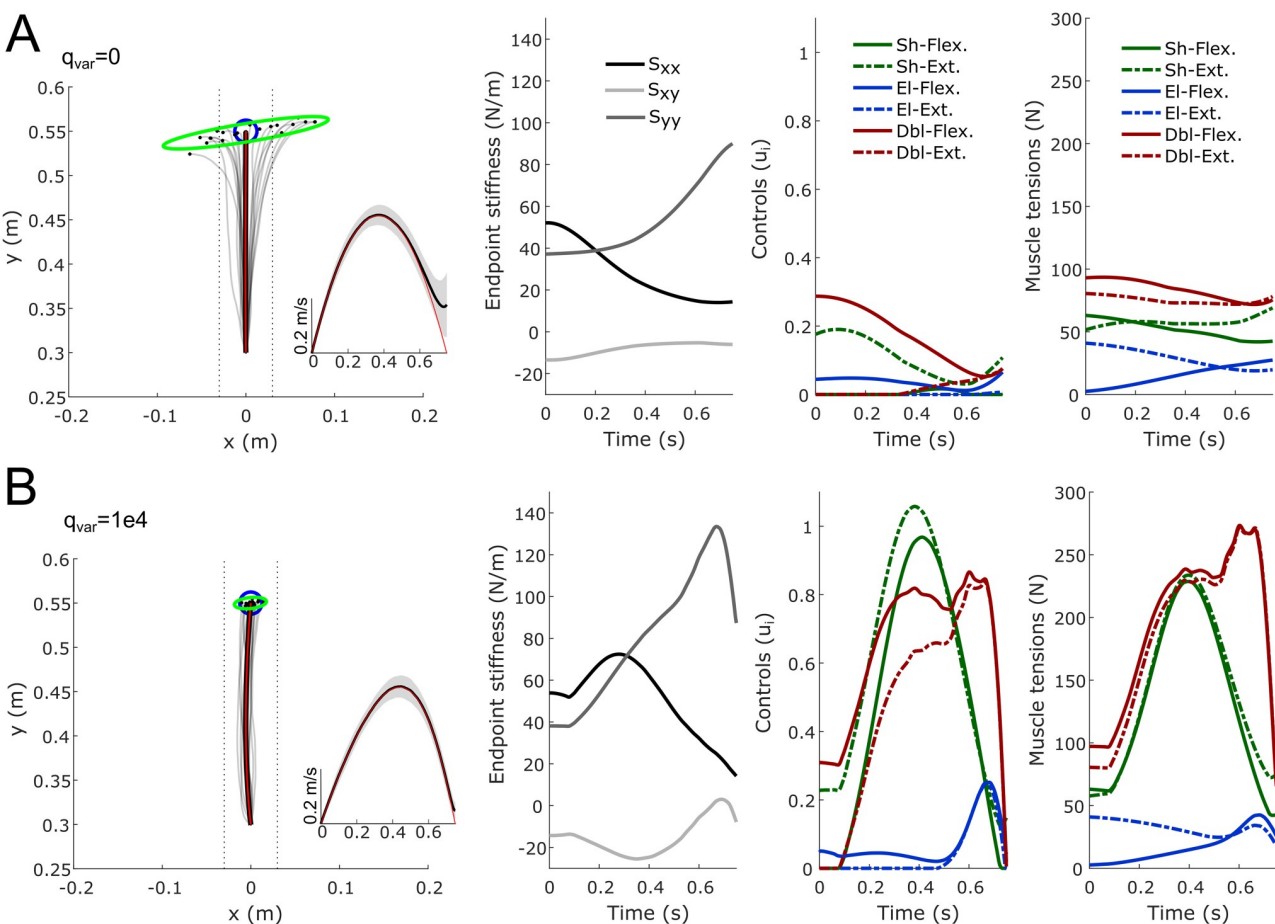

**Fig 4. Two-link arm reaching experiment in a divergent force field.** A. Endpoint trajectories (paths and velocity) and stiffness (Cartesian stiffness components $S_{xx}$, $S_{yy}$ and $S_{xy}$ of the matrix $S = J^{-\top}(\mathbf{q})KJ^{-1}(\mathbf{q})$ where $K = \frac{\partial \tau}{\partial \mathbf{q}}$ is the joint stiffness) when no penalization of the endpoint variance is considered in the cost ($q_{var} = 0$). Twenty sampled paths of the endpoint are depicted (light gray traces). Red traces represent the theoretical mean trajectory from the associated DOC problem and thick black traces represent the mean trajectory over 1000 samples. Vertical dotted lines are displayed to visualize deviations from the straight path (±3cm). The blue circle represents the target (radius of 2.5cm). The green ellipse represents the endpoint positional covariance. The temporal evolution of the mean endpoint stiffness is also depicted (components $S_{xx}$, $S_{yy}$ and $S_{xy}$). The six muscle activations (open-loop control variables) and the muscle tensions are also reported. Muscles with opposed biomechanical functions were paired (emphasized with the same color, solid lines for flexors and dashed lines for extensors). B. Same information with a weight on the variance cost equal to $q_{var} = 10^4$. A significant increase of co-contraction of agonist/antagonist muscles can be noticed and the improvement in final accuracy is also clear (green ellipse). Note that only an open-loop motor command was employed in these simulations (no on-line state feedback control to correct deviations due to noise and the divergent force field).

However, while endpoint stiffness increased along the direction of instability, we also found that it increased in the direction of the movement. As such, the stiffness geometry was not really shaped according to the direction of the destabilizing force (see Fig 5A). We found this is actually a limitation of the underlying musculoskeletal model used in these simulations, which does not allow arbitrary geometries of the endpoint stiffness for the current arm posture –the orientation of the depicted ellipses was actually the most horizontal that one can get from this model given the Jacobian matrix at the midpoint of the trajectory–. This was tested by considering all possible muscle activation vectors $\mathbf{u} \in \mathbb{R}_+^6$ and checking that the resultant stiffness ellipse was never oriented horizontally. To increase the stiffness along the x-axis, the algorithm thus had to increase the endpoint stiffness as a whole and not as selectively as in the data of [16, 23] (but see [71] in static tasks). Note that this observation does not preclude alternative

 

**Table 2. Influence of model parameters on the simulated optimal movements.** The model parameters that were varied are $\sigma_i$, $q_{var}$ and $\beta$. Effort is the integral cost in the control variable $\mathbf{u}(t)$. EP std is the final standard deviation of the endpoint along the x-axis, and $\bar{S}_{xx}$ is the mean endpoint stiffness along the x-axis. Sensitivity of the results to increasing the magnitude of the force field, increasing noise or increasing $q_{var}$ is tested. Note that the model predicts an increase of the lateral endpoint stiffness on average to perform the task accurately in open-loop (~2x factor in these simulations).

| Noise $\sigma_i$ | $q_{var}$ | $\beta$ (N/m) | Effort ($\times 10^{-2}$) | EP std (cm) | $\bar{S}_{xx}$ (N/m) |
|---|---|---|---|---|---|
| 0.025 | 0 | 0 | 3.73 | 0.79 | 28.02 |
| 0.025 | 0 | 40 | 3.98 | 4.46 | 28.29 |
| 0.025 | 1e4 | 0 | 18.18 | 0.52 | 32.20 |
| 0.025 | 1e4 | 40 | 120.77 | 0.89 | 50.69 |
| 0.025 | 1e5 | 40 | 306.28 | 0.42 | 60.66 |
| 0.05 | 1e4 | 40 | 213.13 | 1.13 | 57.03 |
| 0.025 | 1e4 | 80 | 356.54 | 1.21 | 81.69 |

musculoskeletal models from yielding stiffness ellipses elongated in the direction of instability (e.g. see [72, 73]). Instead of considering alternative muscle models, we investigated whether the selective tuning of stiffness geometry could be predicted by the SOOC framework by considering a simpler Cartesian model of the task (following the derivations of Eqs 9–13, but for a planar mass-point system). In this Cartesian model, the cost was only composed of effort and endpoint variance terms (no smoothness term was needed because optimal paths were straight when minimizing effort). Using this Cartesian model, it becomes clear that the optimal endpoint stiffness predicted by the SOOC framework is shaped according to the direction of the destabilizing force (see Fig 5B). The interest of this Cartesian model is to show that the change of stiffness geometry in the divergent force field can be explained within the SOOC framework. The interest of the muscle model was to show that muscle co-contraction may indeed underlie the increase of endpoint stiffness.

**A *no* intervention principle.** Finally, we revisit the minimum intervention principle [9] as illustrated in a pointing-to-a-line task in [9, 74, 75]. For this kind of tasks, DOC models will fail to explain the empirical structure of endpoint variability [75]. In contrast, SOC models will capture endpoint variability very well through the minimal intervention principle which states that deviations from the planned trajectory are corrected only when they interfere with the goal of the task [9]. Alternatively a terminal optimal feedback controller can also reproduce this variability but it requires on-line state estimation processes as well [74] (the model re-plans open-loop trajectories from each estimated initial state and is not stochastically optimal in the sense that it does not consider variability across repeated trajectories to determine the control action). Here we show that on-line state estimation through sensory feedback is even not necessary at all to reproduce that variability in task-irrelevant dimensions is larger than variability in task-relevant ones (e.g. uncontrolled manifold, [76]) as long as mechanical impedance is appropriately regulated via feedforward processes like co-contraction (Fig 6).

We considered the same 6-muscle model than in the previous simulation. The simulations show that the endpoint variance is elongated along the target line (i.e. task-irrelevant dimension), showing that impedance regulation can lead to a phenomenon similar to a minimal intervention principle (except that here it should rather be called a no intervention principle as there is no state feedback at all during movement execution). Conceivably, a testable hypothesis to determine whether this consideration makes sense would be to check the presence of such task-dependent endpoint distributions in deafferented patients with no vision (but with initial vision of the arm prior to the movement as in [43] and of the redundant target). It has

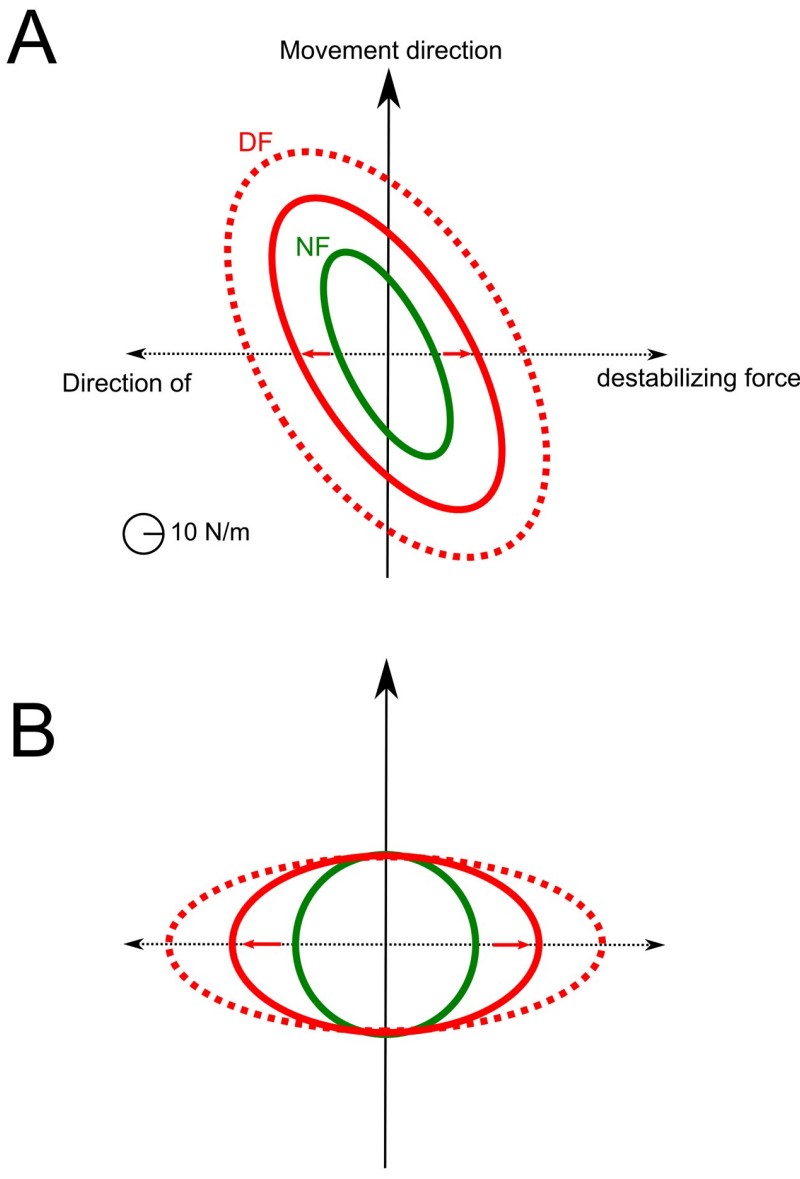

**Fig 5. Endpoint stiffness geometry at the midpoint of movement path.** A. Case of the 6-muscles model with detailed results reported in Fig 4. B. Case of a 2-D Cartesian mass-point model. In green, the geometry of the optimal endpoint stiffness without divergent force field (NF) is represented. In red, the same data is reported when the divergent force field is on (DF). Solid lines correspond to $\beta = 40$ and dotted lines to $\beta = 80$ (note that $q_{var}$ and $\sigma_i$ were fixed in these simulations).

already been shown that, in healthy subjects, this principle still applies when on-line vision is removed (see [75]).

## Discussion

In this study, we have presented a novel optimal control framework to account for the planning of force and impedance via muscle co-contraction. This framework models motor planning as the design of optimal open-loop controls for stochastic dynamics. One main implication is that such open-loop controls will seek to optimally exploit the limb's impedance

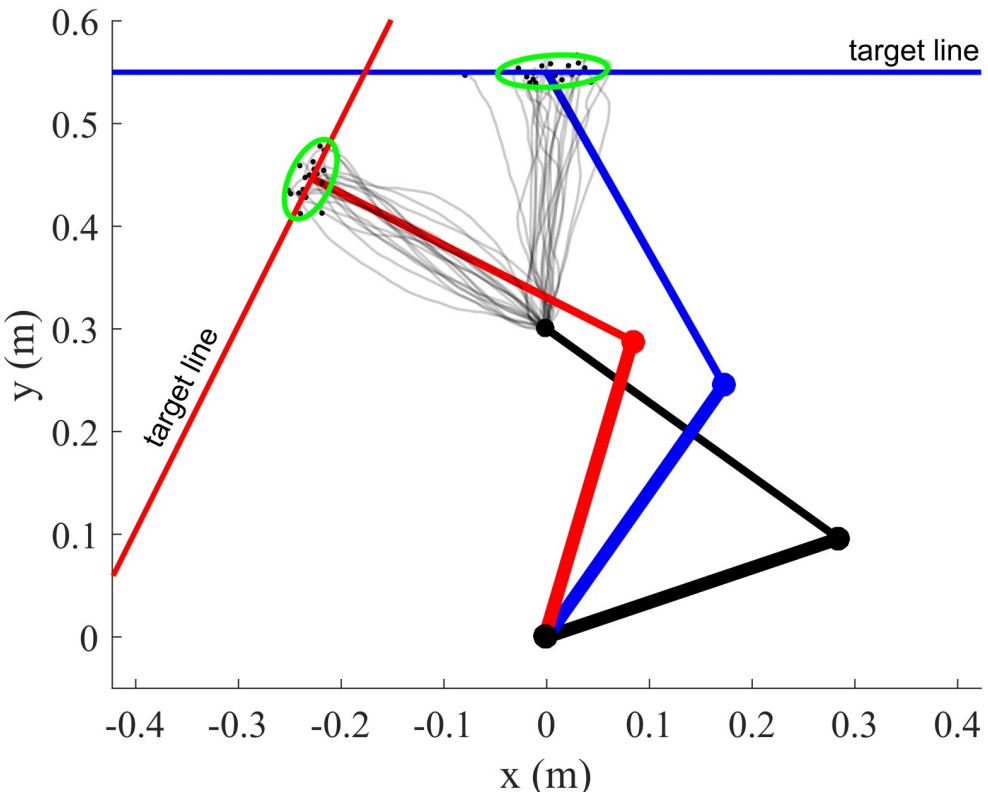

**Fig 6. Top view of arm trajectories for a pointing-to-a-line experiment.** The targets are indicated by the solid lines (blue and red). The green ellipse represents the 90% confidence ellipse of the endpoint distribution. Noise was additive ($\sigma_i = 0.1$) in these simulations and movement time was $t_f = 0.75$s for the forward motion (blue) and $t_f = 0.55$s for the leftward motion (red). The variance weight in the cost, $q_{var}$, was set to $10^4$ and endpoint variance was penalized in a the direction orthogonal to the target line (via the function $\mathbf{n}^\top J(\mathbf{q}_f) P_{\mathbf{q},f} J(\mathbf{q}_f)^\top \mathbf{n}$ where $\mathbf{n}$ is the normal vector, $J$ is the Jacobian matrix and $P_{\mathbf{q},f}$ is the joint-space positional covariance). Note that hard terminal constraints were imposed on the mean state (mean endpoint position on the target line and zero final mean velocity). The main orientation of the endpoint confidence ellipses is compatible with experimental observations and shows that co-contraction may be used to increase accuracy in the task-relevant dimension.

characteristics to perform the task accurately enough, taking into account the presence of uncertainty induced by sensorimotor noise. Optimality is considered with respect to a trade-off between effort and variance costs but other terms may be represented as well (e.g. smooth-ness) in agreement with the literature. Using several simulations, we have illustrated the rele-vance and versatility of the framework to explain well-known experimental observations involving co-contraction and impedance measurements. Below, we discuss the significance and implication of this framework with respect to existing motor control theories.

## Planning of force and impedance via muscle co-contraction

At a computational level, the SOOC framework lies in-between deterministic optimal control and stochastic optimal control theories (see [1, 4, 11] for reviews). These previous frameworks have been useful to predict many aspects of sensorimotor control and learning. However, they usually do not account for the phenomenon of muscle co-contraction in a principled manner. Yet, co-contraction has been found in many motor tasks and is a general feature of motor con-trol (e.g. [17, 18, 48, 77]). In SOOC, the crucial ingredients to obtain co-contraction in muscu-loskeletal systems are threefold: (1) the consideration of open-loop control, (2) the presence of

noise in the dynamics and (3) a cost function including at least effort and variance terms. Each ingredient has found experimental support in the literature. The feedforward aspect of control for learned movements has been emphasized in [18, 38], the effects of sensorimotor noise have been described in [12, 13], and the relevance of effort and variance costs in motor control has been stressed in several studies [65, 78]. The class of models considered in this study is particularly in the spirit of the minimum variance model [49] but with a couple of notable differences. In our framework, effort and variance are separated cost elements such that an optimal motor strategy may involve a large effort without implying a large variance (i.e. co-contraction). In the classical minimum variance model, variance is indeed the same as effort because a signal-dependent noise is assumed to affect a linear system. In our approach, relevant predictions regarding co-contraction and impedance planning can be made only for nonlinear systems (e.g. bilinear systems) and irrespective of the precise type of noise that is modeled (signal-dependent, constant noise etc.). Concretely, the controller can reach different levels of endpoint variance by setting different levels of co-contraction whereas the standard minimum variance model would only yield one (optimal) level of variance (at fixed movement time). Besides variance, effort and energy-like criteria are often minimized in optimal control models which tend to prevent the relevance of co-contraction. In [79], it was demonstrated mathematically that co-contraction of opposing muscles is non-optimal with respect to the minimization of the absolute work of muscle torques in a deterministic model. In other optimal control models with muscle modeling, co-contraction does not occur neither in deterministic settings (Fig. 9 in [26]) nor in stochastic settings (Fig. 2 in [28] or Fig. 3a in [29]). Researchers have nevertheless attempted to explain co-contraction or its contribution to impedance in existing DOC or SOC frameworks, but this was often an *ad-hoc* modeling [80, 81]. One difficulty is that empirical works stressed a relatively complex task-dependency of muscle co-contraction – as assessed by EMG co-activation– [17, 48, 63, 64, 82]. For instance, co-contraction seems to depend on noise magnitude [25, 61] and to tune impedance according to the degree of instability of the task [16, 18, 23, 70, 71]. Finding general principles to automatically predict the adequate co-contraction or impedance required for the task at hand thus appears necessary. In [72, 73, 83], an algorithm focusing on the trial-by-trial learning of force and impedance was developed to acquire stability without superfluous effort. In contrast to this approach, our framework models learned behaviors with known dynamics (up to some uncertainty modeled by noise). Furthermore, while the previous learning algorithm requires a reference trajectory to be defined *a priori* to apply, the SOOC framework allows predicting the "reference" trajectory as the outcome of optimality. This aspect may be particularly important given that inertia is also a key component of mechanical impedance for a multi-joint system. Hence, besides viscoelasticity, the SOOC framework should be able to exploit kinematic redundancy to plan stable behaviors with low effort. Using the SOC framework, other authors have also attempted to predict a limb's mechanical impedance via muscle co-contraction. In [29], a model based on an extended signal-dependent noise model (see Eq 20), which explicitly favors co-contraction by reducing the variance of noise during co-contraction, was proposed. One issue is that for simpler noise models (e.g. simple constant noise), this model would not command muscle co-contraction. In SOOC, co-contraction was planned for a variety of noise models. Co-contraction and stiffness regulation was also considered in another SOC model [84], but the simulated limb's stiffness was mostly due to the intrinsic stiffness of the muscles in the model (that of [53]) without clear task dependency (signal-dependent noise was also a critical constraint). In these SOC models, the role of state feedback could actually duplicate the role of co-contraction –high-level feedback control accounts for a form of impedance but differently from a feedforward co-contraction, [47, 67]–. In [85], an optimal control model was introduced to account for the planning of both trajectory and stiffness. However, this model as well as others

not related to optimal control (e.g. [59, 86, 87]), were derived along the lines of the equilibrium point theory. While co-contraction is often discussed within equilibrium point theory [14], it is worth stressing that our approach rather accounts for co-contraction of a group of muscles within optimal control theory. As such, an important point of departure from equilibrium point theory is the need for internal models of the arm dynamics (see [31]) in order to set net joint torques and derive an efficient feedforward strategy for the control force and impedance in SOOC. The present theory indeed proposes that force and impedance may be planned simultaneously within descending motor commands. This idea seems coherent with the several studies that emphasized that two separate control mechanisms may exist for the control of force and impedance, the latter being at least partly governed by muscle co-contraction [18, 19, 58, 59]. However, impedance can also be regulated via feedback gains in SOC as mentioned earlier and, therefore, the conceptual differences between SOOC and SOC should be discussed further.

## Implications as a motor control theory

Our framework partly formulates motor planning as a stochastic optimal open-loop control problem. One primary outcome of the planning process would then be a feedforward motor command that optimally predetermines both the mean behaviour and the variability of the system via force and impedance control. The term "open-loop" may raise questions about the role of sensory feedback in this framework. Computationally, sensory feedback is only required to estimate the initial state of the motor apparatus during movement preparation in SOOC. This contrasts with optimal closed-loop control that critically requires an estimate the system state to create the motor command throughout movement execution [9, 11]. Indeed, if an optimal feedback gain can be elaborated at the motor planning stage in SOC, the actual motor command is only determined once the current state of the motor apparatus has been properly estimated at the execution stage (e.g. hand or joint position/velocities. . .). An optimal closed-loop control scheme is thought to involve the primary motor cortex and, therefore, to require on-line transcortical feedback loops [32–34]. These neural pathways imply relatively long latencies with muscle responses occurring ~50-100 ms after a mechanical perturbation is applied to a limb. Because these responses are quite sophisticated and task-dependent, relatively complex cortical processes combining sensory information with predictions from internal models of the limb's dynamics and the environment are likely necessary for their formation. Besides these long-latency responses, short-latency responses are also observed <40 ms after a mechanical perturbation. This stretch reflex only involves the spinal circuitry and has been shown to be relatively fast, simple and stereotypical. Nevertheless, background muscle activity is also known to modify the gain of the stretch reflex likely due to the size-recruitment principle [88]. Co-contraction is therefore a means to increase the apparent mechanical impedance of a joint by increasing the gains of stretch reflexes in opposing muscles (and not only by increasing the intrinsic stiffness). Nonlinear effects occurring during co-contraction have been shown to amplify this increase of the reflex gains beyond what would have been expected by considering each muscle alone [22]. As we do not exclude the contribution of the stretch reflex in SOOC, ambiguity may arise here. Indeed, the stretch reflex relies on sensory information from muscle spindles: hence it does implement a (low-level) feedback control loop. However, we argue that the functioning of the neuromusculoskeletal system with intact reflex circuitry may be well accounted for within the SOOC framework and the "open-loop" control assumption. Indeed, being mainly under the influence of descending motor commands via alpha/gamma motoneurons activity, the short-latency reflex arc plays a crucial role in the apparent spring-like properties of a muscle –which we model– beyond its intrinsic

short-range stiffness [21]. We thus consider that these low-level feedback loops are part of a neuromuscular actuator with variable impedance which is under a (feedforward) control from higher-level centers. As mentioned earlier, mechanical impedance can also be modified via transcortical reflex loops relying on pre-determined feedback gains. However, the associated mechanisms are of a different nature in that they require evolved state estimation processes. It is noteworthy that the same low-level/high-level dichotomy applies to robotics (as does the SOOC framework actually). Moreover, experimental estimations of a limb's impedance during posture or movement are normally unable to rule out the impact of reflexes on measurements (which can be as short as ~20 ms for biceps brachii in humans, [89]). In summary, the distinction between short-latency/low-level spinal loops and long-latency/high-level transcortical loops parallels the distinction between optimal open-loop and feedback control frameworks in computational terms. The crucial difference between SOC and SOOC theories thus regards the involvement or not of high-level state estimation processes in the on-line control mechanisms. One implication of the SOOC theory is that such sophisticated high-level feedback processes occurring during movement execution may not necessarily be critical to ensure reliable motor performance in well-learned motor behaviors (but this achievement may require muscle co-contraction to some extent).

## Conclusion

A new theoretical framework to model human movement planning has been presented. It provides a specific emphasis on the elaboration of optimal feedforward motor commands for the control of noisy neuromusculoskeletal systems. Interestingly, optimal open-loop strategies spontaneously exhibit co-contraction to generate robust motor behaviors without relying on sophisticated feedback mechanisms that requires state estimation processes during movement execution. In this framework, the magnitude of co-contraction or joint/endpoint stiffness is kept as small as possible because effort is penalized. Yet, depending on the task constraints (e.g. instability, accuracy demand) and uncertainty (e.g. internal and/or external noise magnitudes), a significant feedforward co-contraction or stiffening of the joints/hand may become the optimal strategy. This prediction was very consistent as we found it for both joint-level and muscle-level descriptions of the musculoskeletal dynamics as well as for various noise models. The SOOC framework may thus complement SOC in the following sense: once a motor plan is elaborated, locally optimal feedback control strategies may be designed after linearization around the optimal mean trajectory and open-loop control. One general advantage of planning force and impedance via co-contraction could be to provide a nominal robustness to the system, thereby resisting small perturbations without the need for a continuous multi-sensory integration (e.g. merging of visual and somatosensory information at cortical levels) to optimally estimate the state of the system during movement execution. Adequately tuning muscle co-contraction (even to small levels) might allow the system to be less sensitive to delays, noise and task uncertainty (and might improve the reliability of state estimation as well). This theoretical framework should be tested more extensively in the future to see whether it constitutes a viable theory for the neural control of movement but it already provides an interesting conceptual trade-off between purely deterministic approaches and purely stochastic approaches. As far as motor planning is of concern and the elaboration of feedforward motor commands is thought to be a significant component of the neural control of movement (see also [90]), the SOOC theory may constitute a relevant theoretical approach. Finally, we note that the very same framework could prove useful in human-inspired robotics especially for robots with variable impedance actuators [47, 91].

## Acknowledgments

The authors would like to thank Dr. Francesco Nori for having initiated this work during a visiting professorship. The authors also wish to thank Prof. Etienne Burdet for his insightful comments on a previous version of the article.

## Author Contributions

**Conceptualization:** Bastien Berret, Frédéric Jean.

**Formal analysis:** Bastien Berret, Frédéric Jean.

**Methodology:** Bastien Berret, Frédéric Jean.

**Software:** Bastien Berret.

**Writing – original draft:** Bastien Berret.

**Writing – review & editing:** Bastien Berret, Frédéric Jean.

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
