## [Decision Letter · Decision Letter 0]

29 Oct 2019

Dear Dr Berret,

Thank you very much for submitting your manuscript 'Stochastic optimal open-loop control as a theory of force and impedance planning via muscle co-contraction' for review by PLOS Computational Biology. Your manuscript has been fully evaluated by the PLOS Computational Biology editorial team and in this case also by independent peer reviewers. The reviewers overall agreed that the approach and results constitute an important advance in our understanding of co-contraction during movement. The reviewers did, however, raise a number of concerns, as well as many suggestions for how the manuscript could be further improved. While your manuscript cannot be accepted in its present form, we are willing to consider a revised version in which the issues raised by the reviewers have been adequately addressed. We cannot, of course, promise publication at that time.

Sincerely,

Adrian M Haith

Associate Editor

PLOS Computational Biology

Samuel Gershman

Deputy Editor

PLOS Computational Biology

[LINK]

Reviewer's Responses to Questions

**Comments to the Authors:**

Reviewer #1: This paper addresses an important problem: the modelling of feedforward control necessary to deal with large sensory delays in human sensorimotor control i.e. motion planning. Specifically, the paper develops a method to compute the force and mechanical impedance (or muscles activation) for learned arm movements, considering intrinsic motor noise. The method is illustrated by simulating well selected experimental studies from the literature. The presentation is clear and accurate, and i have no major concern. In the following, i will first give general comments, then list minor suggestions.

General comments:

1. The paper seems to target co-contraction of antagonist muscle pairs, however i) mechanical impedance at a joint arises from the co-contraction of the group of all muscles actuating this joint, ii) i guess the technique developed in this paper is valid for such muscle groups, not only for antagonist muscle pairs. If ii) holds, the paper could be formulated in the more general case i).

2. Muscle viscoelasticity varies with their activation, and the human nervous system (NS) can control muscles to shape the interaction with the environment, i.e. impedance control. This environment may involve stable or unstable dynamics, and contain noise. In the paper's current formulation, only internal motor noise is considered, not environmental noise. This should be corrected.

3. Impedance control is shaped by muscle mechanics, stretch reflexes as well as long-delay reflexes, see [Franklin2007, Franklin2008]. Long-delay reflexes are missing in the corresponding description at lines 27-31. Note that they still come naturally under feedforward motor commands as defined by the authors as "defined prior to movement execution".

4. Mechanical impedance is not just stiffness (see e.g. line 33), but can be expressed as corresponding to viscoelasticity. In fact several studies such as [Milner1993] have demonstrated the ability of the NS to adapt wrist viscosity (thus not just stiffness can be controlled, although stiffness and viscosity will co-vary).

5. To facilitate the reading, would suggest adding a figure 1 to describe the different setups corresponding to equations 1, 14-16, close to these equations and with the related parameters.

6. The proposed model can predict muscles activation of movements learned in various stable and unstable environments. Similarly, the model of [Franklin2008,Tee2010] can predict at least the experiments of figures 1,2,4,5. It is currently just mentioned at line 523, but i think a comparison with the new model should be provided. In my understanding:

- The new model determines the trajectory, force and impedance corresponding to the learned behaviour of a limb with known kinematics and dynamics, in a known dynamical environment.

- On the other hand, the model of [Franklin2008,Tee2010] can learn the force and impedance along a reference trajectory, and does not need a-priori knowledge of the plant and environment dynamics or kinematics. Note that while model is formulated in an ad-hoc way in [Franklin2008,Tee2010], it in fact corresponds to the gradient descent minimisation of error and effort as analysed in [Yang2011].

- I guess that the simulations of reaching forward movements with lateral instability, the conditions are different from the experiment [Burdet2001] and the simulations of [Franklin2008,Tee2010], where the external force drops when the hand deviates laterally more than x centimeters from the straight line (the experiment would be dangerous and tiring to carry out without this). This may explain the different terminal muscle activation in the simulations in Fig.4,5.

7. As pointed out by the authors, the major difference of their model to SOC is that it is feedforward while SOC is "closed-loop". I would stress this difference even more by calling the SOC closed-loop control at every opportunity in the text. (also it would be possible to use SOC corresponding to the control community which invented it rather than SOFC used much later in the computational neuroscience community, and FSOC (i.e. feedforward SOC) could then be used instead of SOOC?)

Minor suggestions:

Abstract

While these approaches have yielded valuable insights about motor control, they typically fail explain a common phenomenon known as muscle co-contraction. Co-contraction of agonist and antagonist muscles contributes to modulate the mechanical impedance of the neuromusculoskeletal system (e.g. joint stiffness) and is thought to be mainly under the influence of descending signals from the brain.

->

While these approaches have yielded valuable insights about motor control, they typically fail in explaining muscle co-contraction. Co-contraction of a group of muscles associated to a motor function (e.g. agonist and antagonist muscles spanning a joint) contributes to modulate the mechanical impedance of the neuromusculoskeletal system (e.g. joint viscoelasticity) and is thought to be mainly under the influence of descending signals from the brain.

Optimal feedback (closed-loop) control, preprogramming feedback gains but requiring on-line state estimation processes through long-latency sensory feedback loops,

->

Optimal closed-loop control, ...

Author summary

to explain the planning of force and impedance (e.g. stiffness)

->

to explain the planning of force and impedance (e.g. viscoelasticity)

A major outcome of this mathematical framework is the explanation of a long-known phenomenon called muscle co-contraction (i.e. the concurrent contraction of opposing muscles).

->

A major outcome of this mathematical framework is the explanation of muscle co-contraction (i.e. the concurrent contraction of a group of muscles involved in a motor function).

line 9

On the other hand, stochastic optimal –feedback– control (SOC or SOFC) theory was developed to account for the

->

On the other hand, stochastic optimal control (SOC) was used to account for the ...

16

The SOFC theory led to a number of valuable predictions among which the minimal intervention principle, stating that errors are corrected on-line only when they affect the goal of the task, is a significant outcome [9].

->

The SOC theory led to a number of valuable predictions among which the minimal intervention principle, stating that errors are corrected on-line only when they affect the goal of the task [9].

19

However, these two prominent approaches have in common that they fail to simply account for a fundamental motor control strategy used by the central nervous system

->

However, both of these approaches fail at accounting for a fundamental motor control strategy ...

21

co-contraction or co-activation of antagonist muscles

->

co-contraction or co-activation of muscles groups

27

This effect does not only result from the summation of intrinsic stiffnesses of opposing muscles [20, 21] but also from nonlinear stretch reflex interaction [22, 23].

->

This effect results both from the summation of intrinsic stiffness of muscles around a common joint [20, 21] and reflexes [22,23,Franklin2007].

33

First, co-contraction contributes to modulate the effective limb’s impedance (e.g. joint stiffness),

->

... the effective limb’s impedance (e.g. joint viscoelasticity),

48

More fundamentally, an optimal feedback control scheme requires

->

More fundamentally, a closed-loop optimal control scheme requires ...

53

This may seem to contrast with the feedforward nature of impedance and co-contraction control that has been stressed in several studies [16, 18, 34–36].

->

This may seem to contrast with the feedforward nature of impedance and co-contraction control [16,18,34,Franklin2013B]

(xxx these papers present direct experimental evidence for the feedforward nature of impedance control)

57

As this ability may be limited in some cases (e.g. unstable task or too fast motion), co-contraction

->

As this ability is limited in some cases (e.g. unpredictable interaction with the environment, unstable task or too fast motion), co-contraction

82

Although we use the term open-loop –in the sense of control theory–we do not necessarily exclude the role of automatic short-latency reflexes that contribute to the spring-like behavior of intact muscles beyond their short-range stiffness.

->

... we do not necessarily exclude the role of reflexes that contribute to the spring-like behavior of intact muscles beyond their short-range stiffness.

90

Our working hypothesis is that both force and impedance are planned

->

Our working hypothesis is that both force and mechanical impedance are planned

93

open-loop controls

->

open-loop control

128

where R, Q and Qf are positive definite and positive semi-definite matrices with appropriate dimensions

respectively.

->

where R, Q are positive definite matrices and and Qf is a positive semi-definite matrix, all of appropriate dimensions.

130

it can be put out of the expectation

->

it can be taken outside the expectation value integral

145

has a nonlinear dynamics

->

has nonlinear dynamics

152

in agreement with the well-known minimum variance model

->

in agreement with the minimum variance model

173

1st order Taylor approximations

->

first order Taylor approximations

lines 200 to 220:

is it necessary to invoke Feldman (thus a physiological hypothesis) here, or would a linearisation do the same job

230

To illustrate an enlightening point, let us focus on horizontal movements now. The system then simplifies as follows:

->

Focusing on horizontal movements, the system then simplifies to:

268

A two degrees-of-freedom (dof) version of the arm with 6 muscles was also considered to simulate planar arm reaching movements, corresponding to the full model of [51].

->

A two degrees-of-freedom (dof) version of the arm with 6 muscles was also considered to simulate planar arm reaching movements. This is exactly the full model described in [51].

273

C is the Coriolis/centripetal term

->

C \\dot{q} is the Coriolis/centripetal term

276

The net joint torque vector was a function

->?

The net joint torque vector is a function

291

In the seminal study of Hogan described above [46],

->

In Hogan's study [46] described above,

302:

i) add: "where the parameters are defined by Eq.1."

ii) what are the units of the parameters in this simulation?

309

in such an unstable posture

->

in the unstable posture

311

(remind that we prevent feedback control)

->

(remind that feedback control is prevented)

314

In the loaded case, the task instability is increased

-> what does this mean? what would be the measure of stability?

Fig.1

the lines are currently hardly visible. To increase the visibility, one could e.g. reduce the position range to e.g. [-3,3] and the velocity range to e.g. [-10,10], and indicate the standard deviation using e.g. a fine dotted line?

348

in order to model that co-contraction does not lead to increased variability

->

in order to model that co-contraction does not lead to increased variability [Burdet2001]

Fig.2 For these simulations, why using different q_var values? could for example all simulations be done with q_var=5000 ?

362

behavior of subjects described in [61]

->

behavior of subjects in this experiment

Fig.3, similar to Fig.1, could the visibility be improved?

397

Therefore, it was impossible for the subjects to predict

->

Because the hand would start with random lateral deviation due to motor noise, it was not possible for the subjects to predict ...

404

(e.g. participants kept co-contracting when the divergent force field was unexpectedly removed) [18, 65–67].

->

[18, Franklin2003B,66]

418

there was a paper by Wolpert around 1996 showing how the visual feedback can lead to deforming the hand trajectory, which may back the use of a jerk term for this simulation

445

We noticed this is actually a limit of the 6-muscle model used in these simulations, which does not allow arbitrary geometries for the endpoint stiffness in a given posture

->

xxx note that the geometry of the 2 link model allows modifying the stiffness ellipse shape and orientation, see e.g. [Tee2010]

similarly the difference to [Tee2010] could be mentioned in lines 450-460.

462

Finally, we revisit the minimum intervention principle [9]. This well-known principle is most simply illustrated in a pointing-to-a-line task as in [9, 69, 70].

->

Finally, we revisit the minimum intervention principle [9] as illustrated in a pointing-to-a-line task in [9, 69, 70].

502

the consideration of open-loop controls

->

the consideration of open-loop control, ...

516

effort and energy-like criteria are often minimized in optimal control models which tends to prevent

->

... which tend to prevent

523

Researchers have nevertheless attempted to explain co-contraction or its contribution to impedance in existing DOC or SOFC frameworks, but this was often an ad-hoc modeling [75,76].

->

xxx i believe this requires more discussion, and a comparison with this paper's results. To note, [75, Franklin2008] are formulated in an ad-hoc way, but the underlying mathematical principle is described in [Yang2011].

[Milner1993] TE Milner and C Cloutier C (1993), Compensation for mechanically unstable loading in voluntary wrist movement. Experimental Brain Research 94(3): 522-32.

[Franklin2007]

DW Franklin, G Liaw, TE Milner, R Osu, E Burdet and Kawato (2007), Endpoint stiffness of the arm is directionally tuned to instability in the environment. Journal of Neuroscience 27(29): 7705-16.

[Franklin2008]

DW Franklin, E Burdet E, KP Tee, R Osu, CM Chew, TE Milner and M Kawato (2008), CNS learns stable, accurate, and efficient movements using a simple algorithm. Journal of Neuroscience 28(44): 11165-73.

[Franklin2003B]

DW Franklin, E Burdet, R Osu, M Kawato, TE Milner (2003), Functional significance of stiffness in adaptation of multijoint arm movements to stable and unstable dynamics. Experimental Brain Research 151(2): 145-57.

[Tee2010]

KP Tee, DW Franklin, M Kawato, TE Milner and E Burdet (2010), Concurrent adaptation of force and impedance in the redundant muscle system. Biological Cybernetics 102(1): 31-44.

[Yang2011]

C Yang, G Ganesh, S Haddadin, S Parusel, A Albu-Schaeffer and E Burdet (2011), Human-like adaptation of force and impedance in stable and unstable interactions. IEEE Transactions on Robotics 27(5): 918-30.

Reviewer #2: This paper proposes a stochastic optimal open-loop control theory which enable to plan the movement and the stiffness of a biological system moved by antagonistic muscles. The core of the work is to show how such a model is able to easily exploit the use of co-contraction to account for task uncertainties/disturbances.

The idea is very interesting, and the paper presents a novel contribution which is worth for consideration. However, I have several comments that I would ask the authors to consider as suggestions to improve their manuscript.

The major drawback that I see in this work is related to the significant variability in the definition of cost functions to implement the approach in the different experimental conditions. I am pretty convinced that there could be a unique definition of a general problem definition/cost function able to “work” in all the conditions. This is also motivated by a biological counterpart; indeed it is pretty unlikely that the human motor control employ different feedforward strategies for different tasks, but rather I would expect an unifying framework (which is one of the main point of strength for the equilibrium point hypothesis). I think that this aspect is at least worth a discussion in the manuscript, together with a clarification on the particular choices in defining the optimisation problems.

Additional comments are provided below, divided in major and minors.

Majors:

- In Fig. 1A, the plot of variance in position and velocity plots is not visible, maybe the authors could try with a different set of colors.

- The description of Fig.1 is a bit confused, I would suggest naming the 4 subplots of subfig A (and the same for B) and refer to those labels in the caption in an ordered way.

- I would expect that, as soon as an equilibrium is reached, the parameters are maintained constant for the whole execution. In the simulations shown in Fig.1, instead, it seems that the optimal solution shows some oscillations in the last 0.5 seconds. Is there a modeling reason for this, or it is related to the optimization itself? I think this is a relevant aspect to discuss.

- In section “Reaching task with the forearm” the authors refer to Fig. 2C to show the effect of trajectory-time on the resulting stiffness. However, this is not completely clear in the figure. I guess the higher trajectory-time the lower overall torque, but explicitly indicating the time dependency of the stiffness would be beneficial. Moreover, it could be really interesting observing and comparing the whole optimal stiffness profile at different trajectory-time values (with a suitable time-scaling to enable the comparison).

- Also, in the simulations of section “Reaching task with the forearm” I observe an oscillation in the optimal impedance at the beginning and at the end of the task, is there a “methodological” reason for this? I would expect instead a steady value (as shown in the reference [54] for humans)

- Fig 4, the plot of velocities is not clear. Please report them in dedicated subfigures. Controls and Muscles Tensions subfigures are not explained in the caption.

- I would include further discussions regarding the following points:

o what happens if the stiffness cartesian matrix is not diagonal?

o what happens if there is an unpredicted interaction with the environment (e.g. a contact with the environment, thus a force in a specific direction)?

o Does this approach scale well with the dimensionality of the problem? E.g. is it possible to generalize to full upper limb models?

o is it possible to model dual arm constraints (e.g. executing a task while holding an object)?

Minors and Typos:

- In abstract: “fail to explain”

- In abstract, the sentence “Optimal feedback (closed loop) control, preprogramming feedback gains but requiring on-line state estimation processes through long-latency sensory feedback loops, may then complement this nominal feedforward motor command to fully determine the limb’s mechanical impedance.” Is too long, I would suggest to rephrase by splitting in two.

- In caption of Fig. 1A, “corresponding individual muscle torques are depicted below (black for the flexor activation and gray for the extensor activation)” shouldn’t be filled and dashed line instead?

- Line 210 --- kd = sqrt(iks)? Is not 1/2

- Line 225 --- Weight factors α, β and qvar can be chosen to adjust the optimal behavior of the system. How do you select these parameters?

- Eq 26 should be followed by a comma and not a dot

Reviewer #3: In the article, the authors use an Optimal Control framework to develop multiple models in different state-spaces (muscle level and joint level) to show that the optimal control principles can be used to explain the co-contraction in human movements. This is an important attempt, as the modelling of co-contraction has not been done in this framework before, and the simulation results have replicated multiple experimental results.

Although this is a very interesting approach I have several concerns.

Major.

The key idea in the joint level model is that there is a reference trajectory, which is controlled by joint torque, and the deviation from this trajectory, which is controlled by the co-contraction. The two parts are separable. In other words, there is the part of the model that deal with the trajectory planning, which is not new when it comes to modelling. This planned trajectory is subjected to system noise, and therefore there is another, one-input controller, where the only control input is stiffness. Is this then a non-trivial result, that such model predicts stiffness control? Would the results still hold, if the control was dependent on the torque too?

Inferring from Fig. 1, numerically the state x and control u are of the similar order of magnitude. However, the selected costs for that model Q and R differ by 3 or 4 orders of magnitude (line 303). This would mean that the effect of activation cost is small compared to the state dependent costs, suggesting that the co-contraction is energetically cheap, which is not the case in humans. Is this correct? What is the relative weighting of these costs? How sensitive are the results to this cost? Normally, the cost parameters are selected so that the effects of the separate modalities are comparable, otherwise why have it as a cost in the first place?

Specific.

The joint-level model description considers the model as open-loop. However, the behaviour contains the corrections from the reference trajectory, which is clearly feedback control. The authors should clarify what they mean by open loop in this case?

Joint level modelling is clearly described and easy to follow. However, the muscle level modelling lacks clarity in definition. Authors provide the equations for the mechanical model behaviour, but the implementation of the controller (at least to me) is unclear and not nearly at the level of the joint level model.

I understand that it is probably beyond the scope of the paper, but I would like to see (at least the discussion) of how such model would extend to the case where the feedback control is available. Would co-contraction still be present?

The 6-muscle model is unable to learn to increase the endpoint stiffness only in the direction of the instability (Figure 5A). However previous muscle-based models have shown that this is possible when considering costs of stability, accuracy and metabolic cost (Franklin et al., J Neuroscience, 2008; Tee et al., Biological Cybernetics, 2010; Kadiallah et al., PLoS ONE, 2012). Is this because of the specific parameters of the muscle/joint model that was used, or specific to the newly developed SOOC model. Could this result from the extreme low cost of co-contraction such that learning the specific endpoint impedance is not necessary/optimal?

**Have all data underlying the figures and results presented in the manuscript been provided?**

Reviewer #1: Yes

Reviewer #2: Yes

Reviewer #3: No: The code for the simulations is only available from the authors on request. According to my understanding this should be freely available online after acceptance of the manuscript.

PLOS authors have the option to publish the peer review history of their article (what does this mean?). If published, this will include your full peer review and any attached files.

Reviewer #1: Yes: Etienne Burdet

Reviewer #2: Yes: GIUSEPPE AVERTA

Reviewer #3: No

---

## [Decision Letter · Decision Letter 1]

17 Dec 2019

Dear Dr Berret,

Thank you very much for submitting your manuscript, 'Stochastic optimal open-loop control as a theory of force and impedance planning via muscle co-contraction', to PLOS Computational Biology. As with all papers submitted to the journal, yours was fully evaluated by the PLOS Computational Biology editorial team, and in this case, by independent peer reviewers. The reviewers were satisfied with the revisions to the paper. However, Reviewer 3 still has some outstanding concerns. Although these comments do not require major revisions, it would helpful to briefly clarify the extent to which the SOOC approach might be applicable in more complex settings (e.g with gravity or for a 2-link arm - see their Q1), and whether the failure to predict endpoint stiffness in unstable environments reflects a limitation of the solution method or a limitation of the underlying model (see their Q3).

We would therefore like to ask you to modify the manuscript according to the review recommendations before we can consider your manuscript for acceptance. Your revisions should address the specific points made by each reviewer and we encourage you to respond to particular issues Please note while forming your response, if your article is accepted, you may have the opportunity to make the peer review history publicly available. The record will include editor decision letters (with reviews) and your responses to reviewer comments. If eligible, we will contact you to opt in or out.raised.

- Supporting Information uploaded as separate files, titled 'Dataset', 'Figure', 'Table', 'Text', 'Protocol', 'Audio', or 'Video'.

We hope to receive your revised manuscript within the next 30 days. If you anticipate any delay in its return, we ask that you let us know the expected resubmission date by email at ploscompbiol@plos.org.

Sincerely,

Adrian M Haith

Associate Editor

PLOS Computational Biology

Samuel Gershman

Deputy Editor

PLOS Computational Biology

[LINK]

Reviewer's Responses to Questions

**Comments to the Authors:**

Reviewer #1: The paper is fine to me and the authors have addressed the previous comments in a suitable way.

Three minor points could be addressed:

1. The response had:

C \\dot{q} is the Coriolis/centripetal term

Answer: Changed.

however at line 275 i still read "C is the Coriolis/centripetal term"

i do not know whether this is an overlook.

2. Reference [47] seems to be incomplete:

47. Berret B, Jean F. Efficient computation of optimal open-loop controls for stochastic systems; 2019.

3. In the discussion at lines 570-605, if i understand correctly, mechanical impedance is still ascribed to intrinsic mechanical properties and spinal reflexes. However in the studies of Franklin et al. 2007, 2008 etc. mechanical impedance seems to stem also from long delay reflexes.

Reviewer #2: The authors addressed all the comments I had on the first version of the manuscript and, I believe, the final version of this work is suitable for publication.

Reviewer #3: The authors have revised the paper and answered several of my concerns, however I feel that three of these issues are not sufficiently addressed in the current revision.

Q1:

Original: The key idea in the joint level model is that there is a reference trajectory, which is controlled by joint torque, and the deviation from this trajectory, which is controlled by the co-contraction. The two parts are separable. In other words, there is the part of the model that deal with the trajectory planning, which is not new when it comes to modelling. This planned trajectory is subjected to system noise, and therefore there is another, one-input controller, where the only control input is stiffness. Is this then a non-trivial result, that such model predicts stiffness control? Would the results still hold, if the control was dependent on the torque too?

Answer: This joint-level model is used for the purpose of illustrating how stiffness control may arise in the proposed framework and for illustrating this theoretical uncoupling of net torque control and stiffness control. However, in more general cases (e.g. a planar 2-dof arm or a 1-dof arm with gravity torque), this result would not hold. However, the same theoretical derivation can still be applied. This was said Line 228 and Lines 243-246.

Response: If this result does not hold for a realistic case where there is gravity then how general is this solution? This feels like an important point that is glossed over in the current manuscript.

Q2:

Original: Inferring from Fig. 1, numerically the state x and control u are of the similar order of magnitude. However, the selected costs for that model Q and R differ by 3 or 4 orders of magnitude (line 303). This would mean that the effect of activation cost is small compared to the state dependent costs, suggesting that the cocontraction is energetically cheap, which is not the case in humans. Is this correct? What is the relative weighting of these costs? How sensitive are the results to this cost? Normally, the cost parameters are selected so that the effects of the separate modalities are comparable, otherwise why have it as a cost in the first place?

Answer: In Figure 1, we plotted angles in degrees but in the mathematical model they would be expressed in radians. Hence the order of magnitude is not really similar (x has values much smaller than u actually; typically, x is <0.05 radian while u>1 Nm). Given that these values are squared and integrated over time, this means that there are several orders of magnitude of difference between the variance magnitude and the effort magnitude. Hence, the weights for R and Q which are indeed adapted to make the two cost components (effort and variance) of comparable magnitude. Regarding the sensitivity of the results with respect to the weights of the cost, this is not very sensitive when the weights are similar orders of magnitude. A fine-tuning of the weights was not needed in these simulations.

Response: The authors response to this is clear, however it would be helpful to mention in the paper that these two cost components are of similar magnitude. As well, as the authors mention that the modelling is done in radians, Figure 1 could also use radians for easier comparison.

Q3

Original: The 6-muscle model is unable to learn to increase the endpoint stiffness only in the direction of the instability (Figure 5A). However previous muscle-based models have shown that this is possible when considering costs of stability, accuracy and metabolic cost (Franklin et al., J Neuroscience, 2008; Tee et al., Biological Cybernetics, 2010; Kadiallah et al., PLoS ONE, 2012). Is this because of the specific parameters of the muscle/joint model that was used, or specific to the newly developed SOOC model. Could this result from the extreme low cost of co-contraction such that learning the specific endpoint impedance is not necessary/optimal?

Answer: It is likely that other muscle models may have been used to change the shape of the stiffness ellipse (as mentioned in the referred works). Here we used the model of Katayama and Kawato (with exactly their parameters). When checking all the possible orientations of the stiffness ellipse in the considered posture, we could not find any muscle activation vector that would make the ellipse horizontal (but note that this was possible for other arm postures). We added a few more words on this issue (Lines 450-460). Finally, please note that co-contraction is always a costly strategy in our models, and it is not negligible compared to the variance cost (thanks to the weights in the cost function).

Response: This still does not address my concern. Is the SOOC model unable to find a solution that matches human subjects, or is it the feature of the kinematic model used. If the authors could implement this SOOC approach on a model that was already shown to produce this effect, we would know where this limitation arises. The fact that the kinematic model can’t produce this result does not show that the SOOC model can produce this. I realize that the Cartesian mass-point model is a partial approach but still does not fully address the issue.

**Have all data underlying the figures and results presented in the manuscript been provided?**

Reviewer #1: Yes

Reviewer #2: None

Reviewer #3: Yes

PLOS authors have the option to publish the peer review history of their article (what does this mean?). If published, this will include your full peer review and any attached files.

Reviewer #1: Yes: Etienne Burdet

Reviewer #2: Yes: Giuseppe Averta

Reviewer #3: No

---

## [Editor Report · Decision Letter 2]

23 Dec 2019

Dear Dr Berret,

We are pleased to inform you that your manuscript 'Stochastic optimal open-loop control as a theory of force and impedance planning via muscle co-contraction' has been provisionally accepted for publication in PLOS Computational Biology.

In the meantime, please log into Editorial Manager at https://www.editorialmanager.com/pcompbiol/, click the "Update My Information" link at the top of the page, and update your user information to ensure an efficient production and billing process.

One of the goals of PLOS is to make science accessible to educators and the public. PLOS staff issue occasional press releases and make early versions of PLOS Computational Biology articles available to science writers and journalists. PLOS staff also collaborate with Communication and Public Information Offices and would be happy to work with the relevant people at your institution or funding agency. If your institution or funding agency is interested in promoting your findings, please ask them to coordinate their releases with PLOS (contact ploscompbiol@plos.org).

Thank you again for supporting Open Access publishing. We look forward to publishing your paper in PLOS Computational Biology.

Sincerely,

Adrian M Haith

Associate Editor

PLOS Computational Biology

Samuel Gershman

Deputy Editor

PLOS Computational Biology

---

## [Editor Report · Acceptance letter]

18 Feb 2020

PCOMPBIOL-D-19-01595R2 

Stochastic optimal open-loop control as a theory of force and impedance planning via muscle co-contraction

Dear Dr Berret,

I am pleased to inform you that your manuscript has been formally accepted for publication in PLOS Computational Biology. Your manuscript is now with our production department and you will be notified of the publication date in due course.

With kind regards,

Laura Mallard
